# Evidence for gene essentiality in *Leishmania* using CRISPR

**Wen-Wei Zhang**[ID]**, Greg Matlashewski**[ID]*

Department of Microbiology and Immunology, McGill University, Montreal, Quebec, Canada

* greg.matlashewski@mcgill.ca

## Abstract

The ability to determine the essentiality of a gene in the protozoan parasite *Leishmania* is important to identify potential targets for intervention and understanding the parasite biology. CRISPR gene editing technology has significantly improved gene targeting efficiency in *Leishmania*. There are two commonly used CRISPR gene targeting methods in *Leishmania*; the stable expression of the gRNA and Cas9 using a plasmid containing a *Leishmania* ribosomal RNA gene promoter (rRNA-P stable protocol) and the T7 RNA polymerase based transient gRNA expression system in promastigotes stably expressing Cas9 (T7 transient protocol). There are distinct advantages with both systems. The T7 transient protocol is excellent for high throughput gene deletions and has been used to successfully delete hundreds of *Leishmania* genes to study mutant phenotypes and several research labs are now using this protocol to target all the genes in *L. mexicana* genome. The rRNA-P stable protocol stably expresses the plasmid derived gRNA and has been used to delete or disrupt single and multicopy *Leishmania* genes, perform single nucleotide changes and provide evidence for gene essentiality by directly observing null mutant promastigotes dying in culture. In this study, the rRNA-P stable protocol was used to target 22 *Leishmania* genes in which null mutants were not generated using the T7 transient protocol. Notably, the rRNA-P stable protocol was able to generate alive null mutants for 8 of the 22 genes. These results demonstrate the rRNA-P stable protocol could be used alone or in combination with the T7 transient protocol to investigate gene essentiality in *Leishmania*.

## Introduction

*Leishmania* protozoa are transmitted by infected sand flies and cause human leishmaniasis resulting in pathologies ranging from self-healing cutaneous lesions to lethal visceral infections [1]. There is no vaccine for human leishmaniasis, and existing treatments rely on drugs with toxic side effects and varying efficacy depending on the type of leishmaniasis. The development of an effective vaccine and new treatments remain the research priorities for control and eventual elimination of leishmaniasis [1]. To identify potential drug targets, develop attenuated vaccine strains and understand the biology of this parasite, it is necessary to determine the essentiality and function of *Leishmania* genes [2].

**Data Availability Statement:** All relevant data are within the manuscript and its Supporting Information files.

**Funding:** The author(s) received no specific funding for this work.

**Competing interests:** The authors have declared that no competing interests exist.

CRISPR (clustered regularly interspaced short palindrome repeats) is a gene-editing technology developed from a bacterial antiviral defense system that has been effectively applied to study genes in eukaryotic cells [3–7]. Cas9, an RNA guided endonuclease and a guide RNA (gRNA) are the two main components of CRISPR gene editing system. The gRNA uses its guide sequence to scan the genome and directs Cas9 nuclease to the specific complimentary DNA target site 3 base pairs upstream from a PAM site such as NGG to generate a double-stranded break (DSB). Repair of the DSB through homology directed repair (HDR) can be used to introduce DNA sequences with desired editing, mutations or selectable markers into the DSB site. Because of its relative simplicity, CRISPR gene editing has been widely used in a variety of organisms including *Leishmania*.

CRISPR gene editing has been adapted for use in *Leishmania* since 2015 and has greatly improved gene targeting efficiency in *Leishmania* [8–11]. However, since *Leishmania* does not have the functional nonhomologous end-joining (NHEJ) pathway due to lack of DNA ligase IV, in the absence of repair templates, *Leishmania* mainly uses Single Strand Annealing (SSA) or Microhomology Meditated End Joining (MMEJ) pathways to repair double strand breaks (DSB), which are not efficient and can lead to co-deletion of the target gene and its adjacent genes [9, 10, 12, 13]. Thus, repair template containing an antibiotic selection marker is often used to improve gene targeting specificity and facilitate isolating CRISPR targeted gene mutant [8, 9, 11].

Two CRISPR based protocols have been commonly used for gene editing in *Leishmania* including the plasmid based stable gRNA and Cas9 expression vector using rRNA gene promoter (rRNA-P stable protocol) [10] and the transient gRNA expression protocol using a T7 promoter (T7 transient protocol) [11]. In the rRNA-P stable protocol, the gRNA coding sequence is inserted into a plasmid vector expressing the Cas9 gene under the control of the *Leishmania* ribosomal RNA promoter and stable transfectants are generated [10, 12]. This stable gRNA and Cas 9 expression protocol has been used to generate gene disruption and deletion mutants, single point mutations, and to tag endogenous genes with or without using a selection marker. In particular, the rRNA-P stable protocol has been used to delete multicopy family genes, to reveal essentiality of various *Leishmania* genes by direct observation of dying null mutant cell clones [9, 10, 12–15], and to generate selection marker free attenuated *Leishmania* mutants as vaccine candidates [16–18]. In addition, the gRNA and Cas9 are expressed in a single CRISPR plasmid in rRNA-P stable protocol, making it convenient to carry out gene targeting in new *Leishmania* strains and clinical isolates [10, 19].

In the T7 transient protocol, a *Leishmania* strain is first engineered to constitutively express the Cas9 endonuclease and T7 polymerase, gene editing is achieved by transfecting this strain with linear DNA encoding the gRNA downstream from the T7 promoter sequence and donor DNA encoding antibiotic selection markers or fluorescent protein tags [11]. The gRNA template and donor DNA can be conveniently prepared by PCR [11]. The T7 transient protocol is more convenient and faster than the stable rRNA-P protocol once the *Leishmania* strain constitutively expressing Cas9 and T7 polymerase has been established, as it bypasses the requirement of construction, transfection and selection for the gRNA/Cas9 expression plasmid. The simultaneous transfection of two antibiotic selection marker donors makes it possible to isolate many of the gene null mutants with no need for cell cloning [11]. Thus far, hundreds of *Leishmania* genes have been successfully targeted with this T7 transient protocol and this has greatly expanded our understanding of *Leishmania* genes and biological systems [20–22].

Given that the gRNAs are transiently expressed in the T7 transient protocol [11], it is possible that Cas9 nuclease may require more time to target multiple copy genes or genes in multiploidy chromosomes even though the gene may be dispensable for *Leishmania* viability. Alternatively, gene null mutants with reduced proliferation may be more difficult to isolate

using the T7 transient protocol due to omission of cell cloning [11, 20–22]. We therefore were interested to investigate whether the stable expression of gRNA may be more efficient at establishing null mutants for multicopy genes and genes which are required for optimal cell proliferation. The rRNA-P stable protocol was used to target 22 *Leishmania* genes that, in previous studies, the T7 transient protocol did not generate the live null mutants [20–22]. Notably, the rRNA-P stable protocol was able to generate live null mutants for 8 of these genes and provide evidence that the remaining 14 genes were essential by observing dying and death of the null mutant promastigote clones in culture. These results suggest that the rRNA-P stable protocol is a useful complement to the T7 transient system for investigating gene essentiality and may be more suitable for targeting multicopy genes.

## Results

### Non-essential genes

Initially, it was necessary to consider which *Leishmania* genes to include in this study. A list of *L. mexicana* and *L. donovani* genes in which null mutants were unable to be generated with the T7 transient protocol were selected as shown in Table 1 [20–22]. Using the rRNA-P stable protocol, we determined that it was likewise not possible to generate alive null mutants for 14 of these genes consistent with these genes being essential for promastigote survival (see S1 Fig and below). It was however possible to generate live null mutants for 8 of these genes with the rRNA-P stable protocol including for example the *L. donovani* LdBPK_100590 and LdBPK_230540 genes that encode hypothetical proteins of unknown function (Fig 1). As the general approach used in this study, *Leishmania* promastigotes were transfected with the pLdsaCN (or pLdCN) plasmid expressing Cas9 nuclease and a Ld100590 gene-specific gRNA followed by selection with G418 as outlined in Fig 1A and detailed in Methods. Resistant promastigotes were subsequently transfected with a linear donor DNA consisting of a PCR product containing the Bleomycin resistance gene with 25 bp homology arms to the Cas9 cleavage site followed by selection in culture with Bleomycin and G418. To prevent overgrowth of partially targeted cells and reduce the time to isolate the gene disruption mutants, the double resistance parasites were cloned in a 96 well plate and the cell proliferation status in each well was monitored every 2–3 days. If the gene to be targeted is non-essential, the cloned promastigotes would continue to proliferate. PCR analysis of 3 null clones shows a complete absence of smaller wildtype Ld100590 gene band with the larger band representing the targeted gene containing the Bleomycin resistance gene (Fig 1B). Likewise, 4 null mutants were generated with the rRNA-P stable protocol for the single copy *L. donovani* gene Ld230540 (Fig 1C).

### Essential genes

To understand *Leishmania* biology and identify potential new drug targets, it is necessary to determine the essentiality of *Leishmania* genes [2]. To illustrate how gene essentiality is normally determined with the rRNA-P stable protocol, we use targeting AGC essential kinase 1 gene (LmxM.25.2340, AEK1) as an example (Fig 2). As shown in Table 1, null mutants were previously not generated for the AGC essential kinase 1 gene (LmxM.25.2340, AEK1) with the T7 transient protocol and this was the same outcome using the rRNA-P stable protocol consistent with this being an essential gene. As described above, promastigotes were transfected with pLdsaCN (or pLdCN) plasmid and selected with G418 followed by transfection of the Puromycin resistance gene donor. Ten to fourteen days later, the G418 and Puromycin double resistance parasites were cloned into a 96 well plate. Different from the T7 transient protocol, the constantly expressed Cas9/gRNA complex would continue to scan the genome and target the remaining copy or copies of the gene if still available after cloning. If the gene is required for

**Table 1. Comparison of two CRISPR methods in targeting *Leishmania* genes.**

| Gene ID[1] | Gene Function | Chromosome | Gene copies[2] | T7 transient protocol | | rRNA-P stable protocol | | Required for viability |
|---|---|---|---|---|---|---|---|---|
| | | | | Ref | WT gene Retention[3] | WT gene Retention[4] | Observation of dying cells[5] | |
| LmxM.02.0290 | Mitogen-activated kinase kinase kinase | 2 | 2 | 21 | Yes | Yes | Yes | Yes |
| LmxM.03.0780 | Serine/threonine-protein kinase | 3 | 2 | 21 | Yes | Yes | Yes | Yes |
| LmxM.08.0530 | Protein kinase | 8 | 2 | 21 | Yes | Yes | Yes | Yes |
| LmxM.08_29.1330 | Serine/threonine-protein kinase Aurora kinase 2, AUK2 | 8 | 2 | 21 | Yes | Yes | Yes | Yes |
| LmxM.09.0910 | Calmodulin | 9 | 6 | 20 | Yes | Yes | Yes | Yes |
| LmxM.16.1550 | Component of motile flagella 6 (CMF6) | 16 | 3 | 20 | Yes | No | No | No |
| LmxM.17.0790 | Polo-like protein kinase, PLK | 17 | 2 | 21 | Yes | Yes | Yes | Yes |
| LmxM.20.0960 | Protein kinase | 20 | 2 | 21 | Yes | Yes | Yes | Yes |
| LmxM.20.1180 | Calpain-like cysteine peptidase | 20 | 2 | 20 | Yes[6] | No | No | No |
| LmxM.24.2010 | Phosphatidylinositol 3-kinase, PI3K | 24 | 2 | 21 | Yes | Yes | Yes | Yes |
| LmxM.25.2340 | AGC essential kinase 1, AEK1 | 25 | 2 | 21 | Yes | Yes | Yes | Yes |
| LmxM.30.2860 | Tousled-like kinase, TLK | 30 | 4 | 21 | Yes | Yes | Yes | Yes |
| LmxM.30.2960 | Repressor of differentiation kinase 2, RDK2 | 30 | 4 | 21 | Yes | Yes | Yes | Yes |
| LmxM.34.3960 | Protein kinase A catalytic subunit isoform 2, PKAC2 | 34 | 2 | 21 | Yes[6] | No | No | No |
| LmxM.34.4010 | Protein kinase A catalytic subunit isoform 1, PKAC1 | 34 | 2 | 20,21 | Yes[6] | No | No | No |
| LdBPK_100590 | Hypothetical protein | 10 | 2 | 22 | Yes | No | Yes[7] | No |
| LdBPK_111030 | Hypothetical protein | 11 | 2 | 22 | Yes | Yes | Yes | Yes |
| LdBPK_230540 | Hypothetical protein | 23 | 2 | 22 | Yes | No | Yes[7] | No |
| LdBPK_260650 | Protein of unknown function, DUF2012 | 26 | 2 | 22 | Yes | Yes | Yes | Yes |
| LdBPK_310120 | FG-GAP repeat protein | 31 | 4 | 22 | Yes | No | Yes[7] | No |
| LdBPK_312380 | 3'-nucleotidase/ nuclease | 31 | 4 | 22 | Yes | No | No | No |
| LdBPK_354780 | Hsp70 protein | 35 | 2 | 22 | Yes | Yes | Yes | Yes |

1. LmxM stands for *L. mexicama*; LdBPK stands for *L. donovani*.

2. These numbers do not include the additional gene copies present in the extra chromosome circles if exist.

3. Whether a wild type (WT) gene band was detected in the surviving transfectant population.

4. Whether a wild type (WT) gene band was detected in all surviving transfectants after cloning in a 96 well plate.

5. The dying and dead cells were caused by the disruption of all wild type gene allele present in these clones.

6. Likely because the primers used to verify the CRISPR deletion mutants were non-specific to the target gene, the WT size PCR band detected could as well be derived from the conserved sequence which are also present in the nearby gene (See S2 and S3 Figs). Indeed, Fochler et al have recently shown that LmxM.34.3960 and LmxM.34.4010 alive null mutants could also be generated with T7 transient protocol when the gene-specific primers were used to confirm the CRISPR deletion mutants [26].

7. Likely due to failure of quick adaptation, some dead cells could be observed from the transfectants when these important but non-essential genes were targeted by CRISPR.

viability following single cell cloning, once the remaining copy of the gene has been disrupted by CRISPR, promastigotes would stop growing or multiply slowly to form clumps until the gene products (mRNA and protein) are diluted and degraded to the minimum level required for survival, as shown for the LmxM.25.2340 null mutants in Fig 2A. Depending on the relative importance, initial abundance, stability of the gene product in the cell and the cloning time (stage) after the complete gene disruption for the individual clone, the dying (dead) cell clumps could contain only a few promastigotes to more than hundreds of promastigotes (Fig 2A,

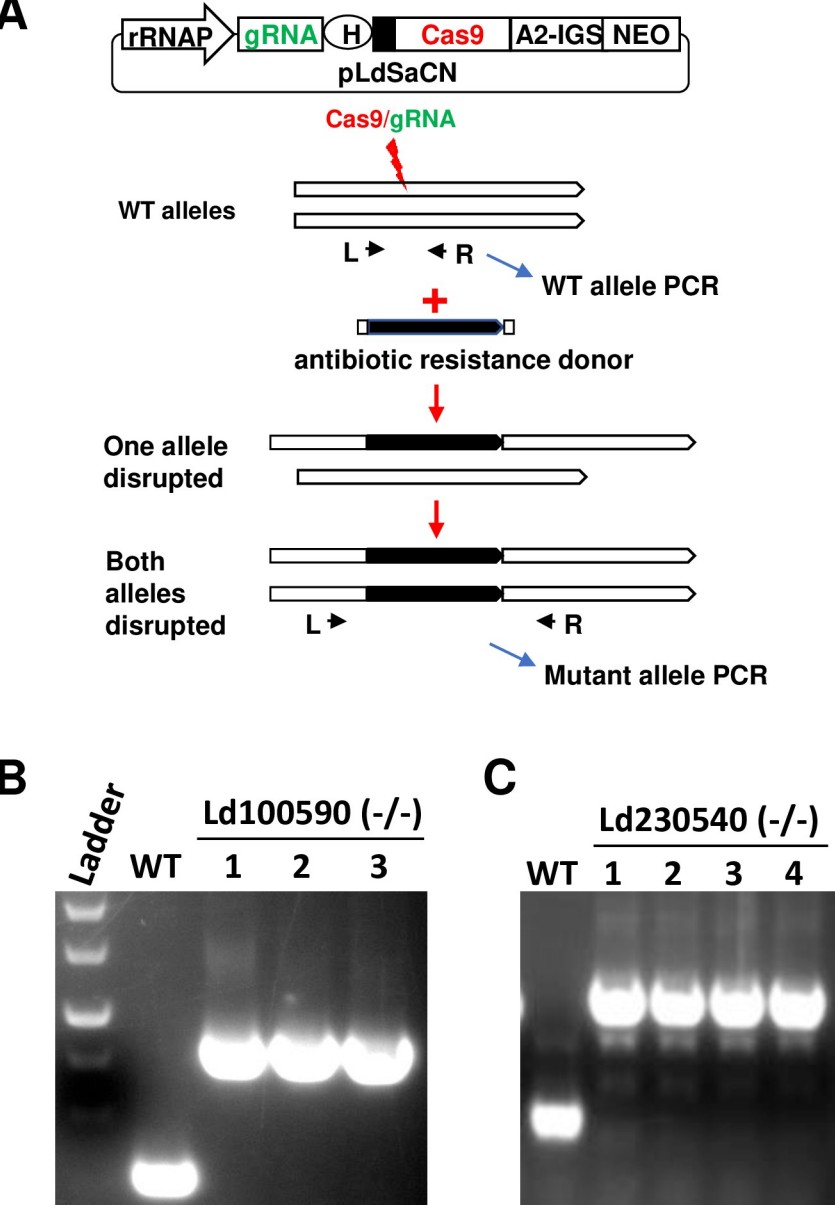

**Fig 1. CRISPR gene targeting strategy (rRNA-P stable protocol) used to disrupt the essential and non-essential *Leishmania* genes in this study.** (A) Schematics of *Leishmania* CRISPR plasmid pLdSaCN and strategy used for gene disruption. In pLdSaCN CRISPR plasmid, *L. donovani* rRNA promoter (rRNAP) controls the stable transcription of the gRNA and *Staphylococcus aureus* Cas9 (SaCas9). The target gene specific gRNA leads Cas9 to generate a double strand break which is then repaired following the introduction of the donor DNA (black bars) containing an antibiotic resistance gene, resulting in disruption of the target gene. A2-IGS, A2 intergenic sequence; NEO, Neomycin resistance gene. Note, the pLdSaCN plasmid is structurally like the other stable expression *Leishmania* CRISPR plasmid pLdCN also used in this study. Instead, pLdCN expresses the commonly used *Streptococcus pyogenes* Cas9 (SpCas9). (B) PCR analysis with primers L and R showing the disruption of a non-essential *L. donovani* gene (LdBPK_100590). The higher disrupted gene band with the inserted donor DNA but not the lower WT gene band was detected in three Ld100590(-/-) clones. (C) Complete disruption of a non-essential *L. donovani* gene (LdBPK_230540).

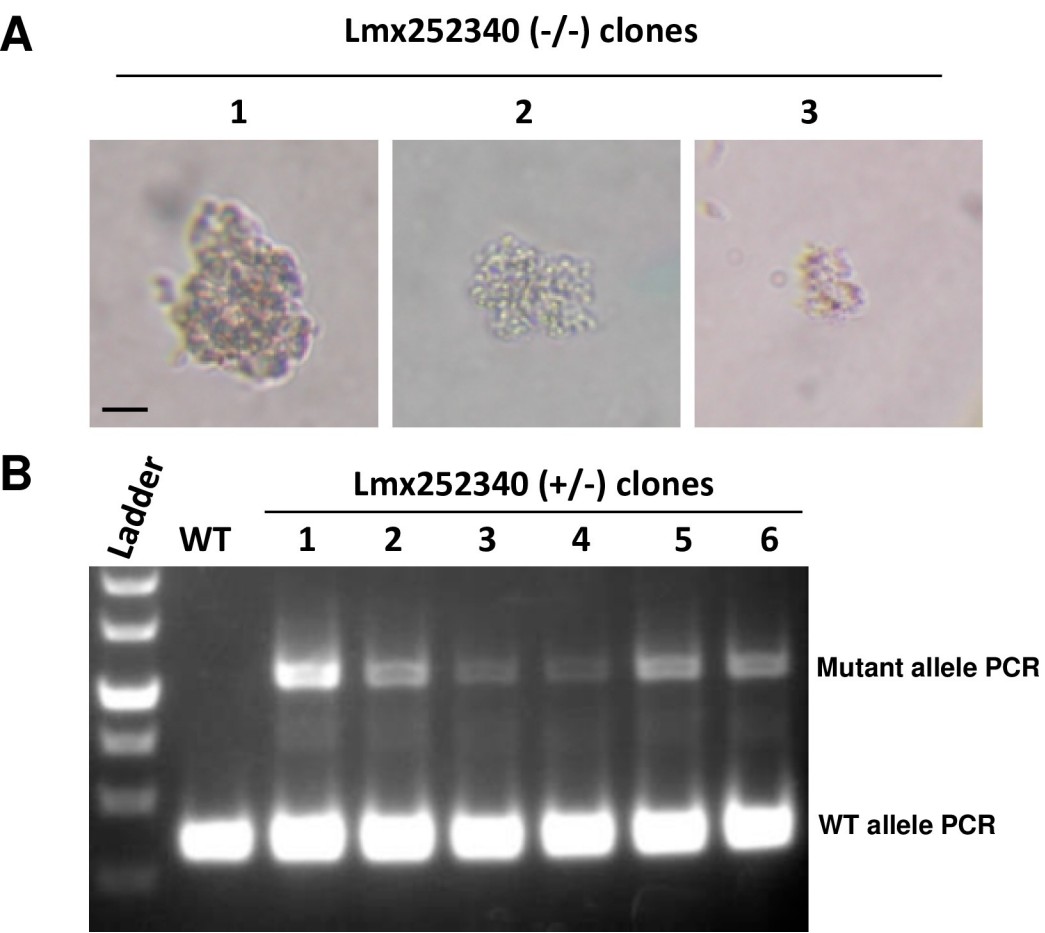

**Fig 2. Evidence the AGC essential kinase 1 gene (LmxM.25.2340, AEK1) is essential for *L. mexicana*.** (A) The dying promastigotes clumps observed in some of the 96 well plate wells after the G418 and Puromycin double resistance transfectants were cloned into a 96 well plate. Scale bar, 15 µm. (B) PCR analysis showing the wild type (WT) AEK1 gene band persists in all surviving clones in the 96 well plate and at least one of the gene alleles was correctly disrupted by CRISPR.

S1 Fig and below). PCR analysis of all the surviving clones in the 96 well plate will show the WT gene band persists and at least one allele of the essential gene was successfully targeted and disrupted by CRISPR (Fig 2B, S1 Fig and below) [9, 10, 12, 14]. In this manner, by combining observation of the death of gene null mutant clones and detection of the WT gene band and the gene targeting band in all surviving clones, using the rRNA-P stable CRISPR protocol, we were able to confirm that many of those genes for which the T7 transient protocol was not able to generate alive null mutants (14 out of 22; see Table 1, Figs 2, 3, and S1 Fig) are truly essential for *Leishmania* viability.

As shown in Table 1, the rRNA-P stable protocol and the T7 transient protocol provide evidence that the LmxM.09.0910 encoding a calmodulin analog is an essential gene. Calmodulin is a highly conserved $Ca^{2+}$ binding protein present in all eukaryotic cells that relays signals to various calcium-sensitive enzymes, ion channels and other proteins. *Leishmania* contains 3 copies of *calmodulin* gene (LmxM.09.0910, LmxM.09.0920 and LmxM.09.0930) in tandem array in chromosome 9 (Fig 3A). The strategy used by the rRNA-P stable protocol to target the *L. mexicana* calmodulin gene family and one of its outcomes are illustrated in Fig 3B. The pLdsaCN plasmid expressing a single gRNA targeting all 3 *calmodulin* genes was transfected

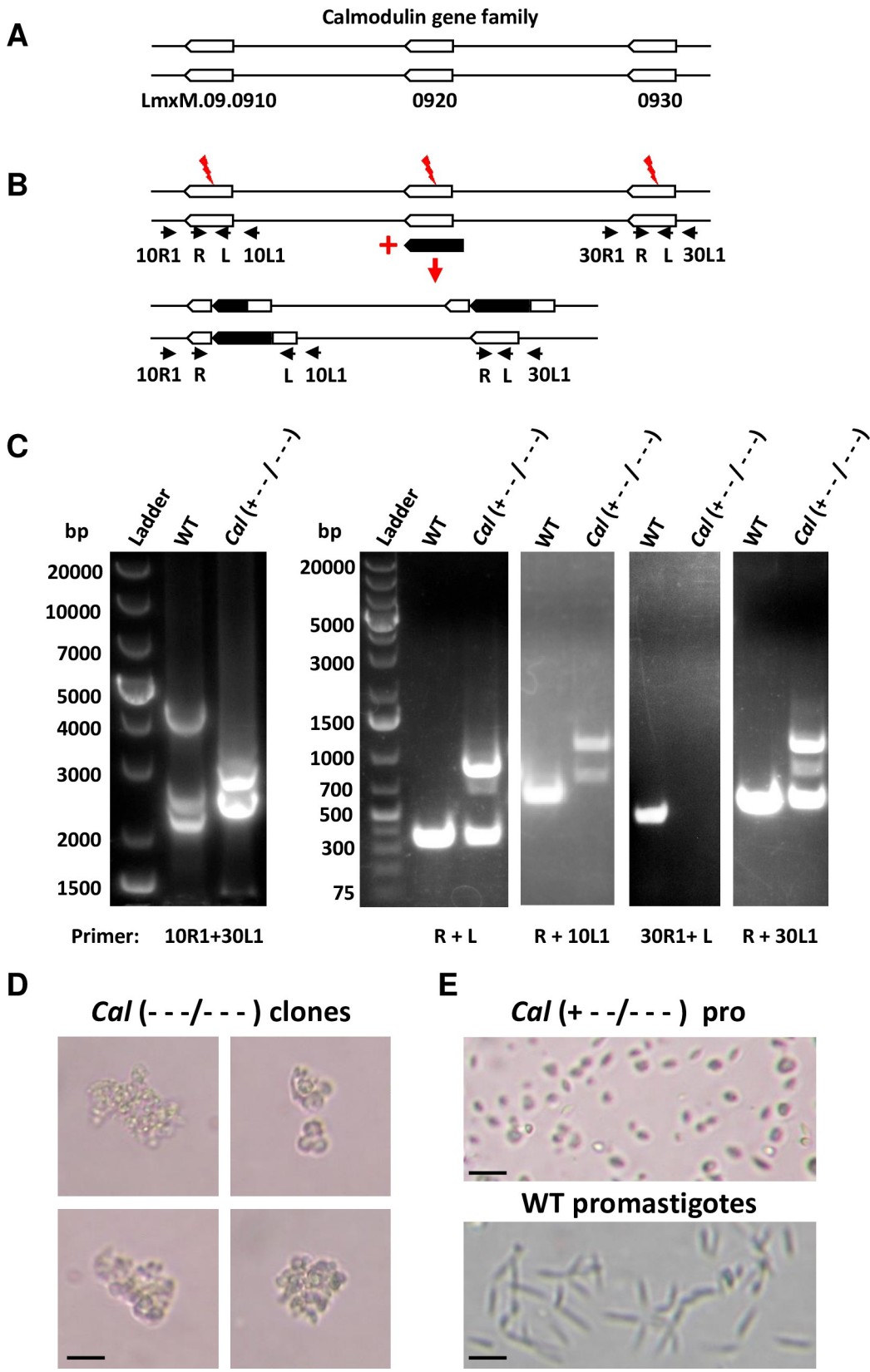

**A** Calmodulin gene family

LmxM.09.0910    0920    0930

**B** 10R1 R L 10L1    +    30R1 R L 30L1

10R1 R    L 10L1    R L 30L1

**C**

Primer:    10R1+30L1    R + L    R + 10L1    30R1+ L    R + 30L1

**D** *Cal* (- - -/- - - ) clones

**E** *Cal* (+ - -/- - - )  pro

WT promastigotes

**Fig 3. Evidence the *Calmodulin* gene is essential for *L. mexicana*.** (A) *Calmodulin* gene locus in chromosome 9 contains three *Calmodulin* genes (LmxM.09.0910; 0920 and 0930) in tandem array. (B) Strategy used by rRNA-P stable protocol to delete and disrupt *L. mexicana Calmodulin* gene clusters, and the outcome of one of the *Calmodulin* gene targeting clones. The PCR primers used to verify gene deletion and disruption are indicated. (C) PCR analysis of the *Calmodulin* gene targeting clone shown in B. After CRISPR gene targeting, the PCR product size with primers 10R1+30L1 reduced from 4.2 Kb of the WT band size to 2.8 Kb and no PCR product could be detected with primers 30R1+L, indicating LmxM.09.0930 gene had been deleted. The PCR product sizes of primers R+10 L1 increased from 700 bp to 900 bp and 1100 bp respectively, indicating both LmxM.09.0910 alleles had been disrupted. Both the WT band and size increased disruption band of primers R+30L1 were detected, indicating one LmxM.09.0920 allele had been successfully disrupted and one WT LmxM.09.0920 allele (the only WT *Calmodulin* gene allele) remained in this clone. Note: a smaller than the expected size (R+10R1) band (900 bp) was detected in this *Cal* (+—/—) clone, indicating a recombination deletion had occurred in one of the two disrupted LmxM 09.0910 alleles, which explains the additional fainter band running at 700 bp detected in the PCR with (R+L) primer pair. Likewise, the fainter band (the middle band) detected in the PCR with (R+30L1) primer pair could be derived from some of the cells in this *Cal* (+—/—) clone cell population where recombination deletion took place in the disrupted LmxM 09.0920 allele. (D) The dead cell clumps observed after the CRISPR gene targeting transfectants were cloned into a 96 well plate, suggesting all the *Calmodulin* genes in the dead clones had been deleted or disrupted. (E) Compared with the WT promastigotes, one of the surviving *Calmodulin* CRISPR gene targeting clones (+—/—) in the 96 well plate is shown with a smaller and round morphology. Scale bar, 15 μm.

followed by the Bleomycin resistance gene donor. As expected, dying (dead) clones were observed in some wells after the transfectants were cloned (Fig 3D and S1 Fig). PCR analysis showed that at least one copy of the wild type of *Calmodulin* gene was retained in all surviving clones (S1 Fig). Further analysis of the slowest growing clone revealed (see detailed explanation in Fig 3 legend) CRISPR gene targeting had deleted both alleles of LmxM.09.0930, disrupted both alleles of LmxM.09.0910 and one allele of LmxM.09.0920, and only one wild type LmxM.09.0920 allele remained (Fig 3B and 3C). The promastigotes of the slowest growing clones were less motile and more rounded than wild-type promastigotes (Fig 3E and below). These data are consistent with the previous report [20] providing evidence that like in other organisms, Calmodulin is essential for *Leishmania*. It is also interesting to point out that *Calmodulin* is the only one of the 98 *Leishmania* flagellar protein genes which is required for *Leishmania* viability [20].

## Genes on polyploid chromosomes

Genes present on Chromosome 31 have at least 4 copies since this chromosome is tetraploid. For example, the LdBPK_310120 (FG-GAP repeat protein) and the LdBPK_312380 (3'NT/NU) genes are located in chromosome 31 [22]. As indicated in Table 1, it was not possible to delete all the copies of these genes using the T7 transient protocol. We attempted to generate null mutants in these genes using the rRNA-P stable protocol as described above. As shown in Fig 4A, expression of a single gRNA followed by introducing a donor DNA encoding the Bleomycin resistance gene generated clones with the wildtype gene replaced with the larger migrating genes containing the donor DNA and no remaining wildtype gene. This demonstrates that the LdBPK_310120 and LdBPK_312380 genes could be disrupted with constant expression of gRNA and Cas9 to generate null mutants.

In another example of a non-essential gene, the *L. mexicana* LmxM.16.1550 gene (Component of motile flagella 6) (Table 1) is in chromosome 16 which is trisomic in *L. mexicana*. As shown in Fig 4B, it was possible to generate 2 null clones (-/-) with all three copies of the wild-type LmxM.16.1550 gene disrupted with the integrated donor DNA. There were however more clones displaying partial gene targeting (+/-) suggesting there is pressure to retain this gene. This demonstrates that it is advantageous to perform cell cloning following gene targeting to isolate some homozygous gene mutants with multiple copies or clones that are slower growing.

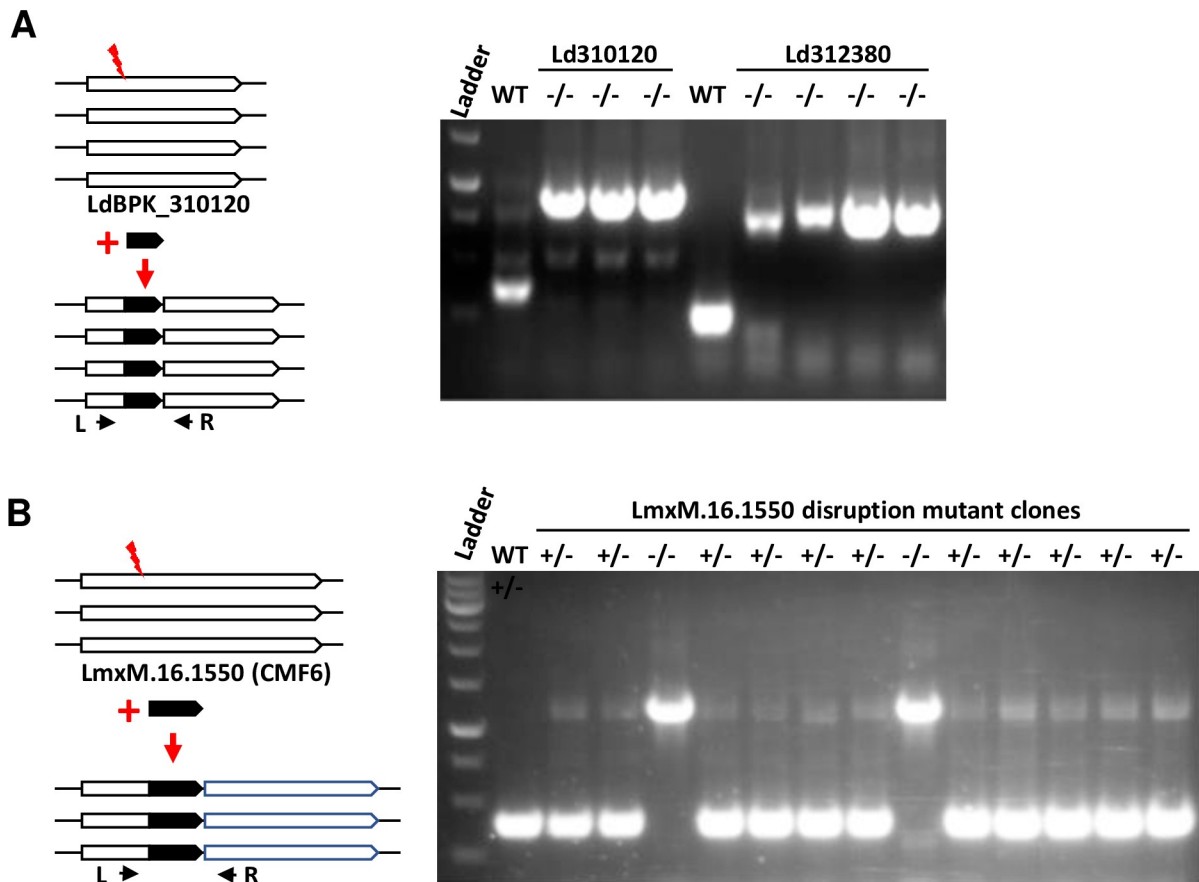

**Fig 4. Disruption of non-essential *Leishmania* genes in polyploid chromosomes with rRNAP-P stable protocol.** (A) CRISPR disruption of *L. donovani* genes LdBPK_310120 and LdBPK_312380 in the tetraploid chromosome 31. A total of four independent Cas9 cleavage and insertion events were required to disrupt all four copies of the genes (see left panel). (B) Disruption of the *L. mexicana* LmxM.16.1550 (CMF6) gene family in the trisomic chromosome 16. Three Cas9 cleavage and insertion events were required to disrupt all three copies of the genes (see left panel). Note, the LmxM.16.1550 gene was completely disrupted in only two of the 13 CRISPR gene targeting clones examined, highlighting the importance of cloning to isolate gene disruption mutants.

## Genes required for optimum proliferation

If a gene product is required for the optimal proliferation of promastigotes, the homozygous gene null mutants could be outcompeted in culture by the heterozygous mutants in the population of CRISPR edited transfectants. This would result in loss of the null mutant unless the mutant was cloned. To investigate the possibility that some null mutants may be slow growing, we compared the proliferation of the null mutants generated in this study. As shown in Fig 5, the proliferation rates of the null mutants LdBPK_100590 and LmxM.16.1550 were much slower than the wildtype promastigotes. Thus, cell cloning would be necessary for isolation of mutants with reduced proliferation such as for example the LmxM.16.1550 mutant clones where a relatively low percentage of null mutants was detected in the above PCR analysis (Fig 4B).

## Phenotypic changes in gene disrupted mutants

Mutation of flagellar protein encoding genes can affect the motility of *L. mexicana* promastigotes in culture [20] and 3 of the *L. mexicana* flagellar genes listed in Table 1 (16.1550, 20.1180, 34.4010) were identified as non-essential using the rRNA-P stable protocol in this

A

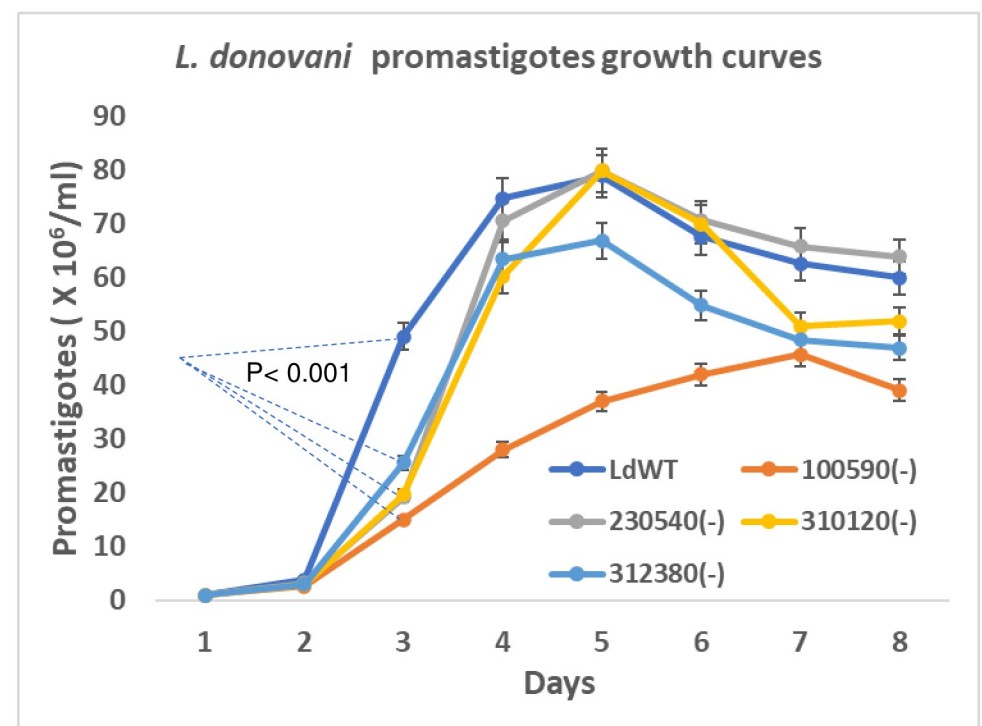

B

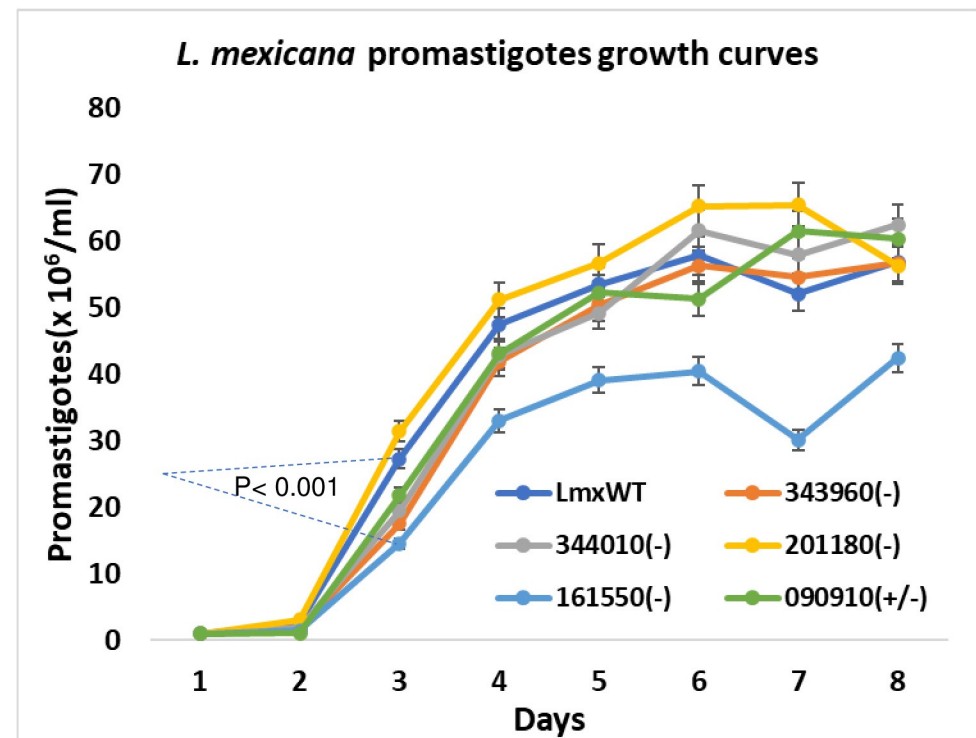

**Fig 5. Promastigote growth curves of the *L.* donovani and *L.* mexicana null mutants generated in this study.** Equal numbers of *Leishmania* promastigotes (1 million promastigotes per ml) were inoculated in flasks each containing 4 ml culture medium. The parasite growth was monitored by microscope counting once a day for 8 days. The data shown are

the mean plus Standard Error of the Mean (SEM). Note, as examples, the cell density differences between LdWT and Ld100590(-), Ld230540(-), Ld310120(-) and Ld312380(-) cells, and between LmxWT and Lmx161550(-) cells at day 3 post inoculation are statistically significant (Student's t-test P<0.001). This is the representative data of three independent experiments.

study. We therefore examined whether the mobility of these mutants was affected using a mobility assay described in Methods. As shown in Fig 6, the LmxM.16.1550 (Component of motile flagella 6) and Lmxm.34.4010 (PKAC1) mutants were largely defective in swimming forward compared to wildtype or the LmxM.20.1180 mutant (S1–S3 Movies). The forward swimming assay showed the number of promastigotes reaching the opposite end of the tube at different time points was significantly reduced for the LmxM.16.1550 and LmxM34.4010 null mutants. Notably, near 50% of LmxM.16.1550 null mutants kept turning around in circles (S2 Movie) despite the flagellar length appeared to be normal (Fig 6B). The LmxM.34.4010 deficient promastigotes in contrast were largely motionless or wiggled in a slow speed despite the flagellar length and promastigote size appearing normal (S3 Movie and Fig 6B). The *Calmodulin* gene targeting mutant [LmxM.09.0910 (+—/—)] retaining at least one wildtype gene was smaller in size and swam more slowly in culture medium (S4 Movie and Fig 6). In comparison, the LmxM.20.1180 (CALP1.1) null mutants are normal in size but appeared to be able to proliferate and swim slightly faster than the wild type cells (S5 Movie and Figs 5 and 6).

## Discussion

It is important to identify essential genes to define biochemical pathways that represent potential intervention targets [2], and this can be performed in *Leishmania* using CRISPR based methodologies. This study demonstrates the ability of the rRNA stable protocol to provide evidence for gene essentiality. The advantage with the T7 transient protocol is that it omits the requirement to clone a gRNA encoding sequence in the pLdCN plasmid and transfection making it more practical for high throughput gene targeting. The importance of performing high throughput gene deletion cannot be overstated since the mutants provide a wealth of information on the biology of *Leishmania*.

More than 5% of *Leishmania* genes have multiple copies present either in aneuploid chromosomes or in tandem array [23, 24], and this study contends that the rRNA stable protocol is more suitable for generating multi-copy gene null mutants. The *Trypanosoma brucei* RNAi library screen study revealed that at least 10% *T. brucei* genes are essential for viability and 25–37% of genes are required for optimal growth under various culture conditions [25]. An advantage of cloning null mutants is the potential to directly visualize dying promastigotes by microscopy such as for example with the *AEK1* null gene mutant (Fig 2A) and the ability to isolate slower growing clones. At least one copy of the wildtype essential *AEK1* gene allele can be detected by PCR in the remaining surviving clones as seen in Fig 2B.

A five-star methodology approach to establish gene essentiality in *Leishmania* has been proposed which we are in general agreement with [2]. In practice however, it is difficult to meet the most stringent criteria using forced plasmid shuffling and DiCre gene deletion that have so far only been applied to a few *Leishmania* genes [2]. Regardless of how essential genes are classified, this study highlights that the rRNA stable protocol should be considered when it is not possible to generate a null mutant with the T7 transient protocol. In addition to providing evidence for gene essentiality, the ability to generate null mutants is necessary for studying differentiation, virulence, pathogenesis and basic biology of the parasite.

During our study, we have also noted that when targeting a single member of a multicopy gene family, it is important to select PCR primers specific to the targeted gene that do not also

**A**

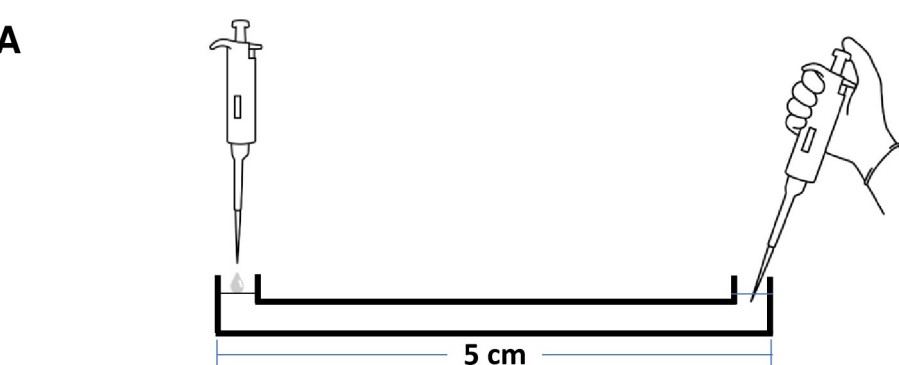

### *L. mexicana* promastigotes detected at the right end of a tube in a swimming assay

| Time (hours) | 1 | 1.5 | 2 | 3 |
|---|---|---|---|---|
| LmxM WT | 14±4 | 58±8 | 230±22 | 439±118 |
| Dead LmxM WT | 0 | 0 | 0 | 0 |
| LmxM.34.4010(-) | 0 | 1.7±0.6 *** | 5.3±1.5 **** | 15±3 ** |
| LmxM.16.1550(-) | 0 | 2.3±0.6 *** | 3±1 **** | 4.3±3 ** |
| LmxM.09.0910(+--/---) | 0 | 11.7±1.5 *** | 14.7±3.2 **** | 135±14 * |
| LmxM.20.1180(-) | 16±5.6 | 71±4.6 | 266±28 | 663±55 * |

**B**

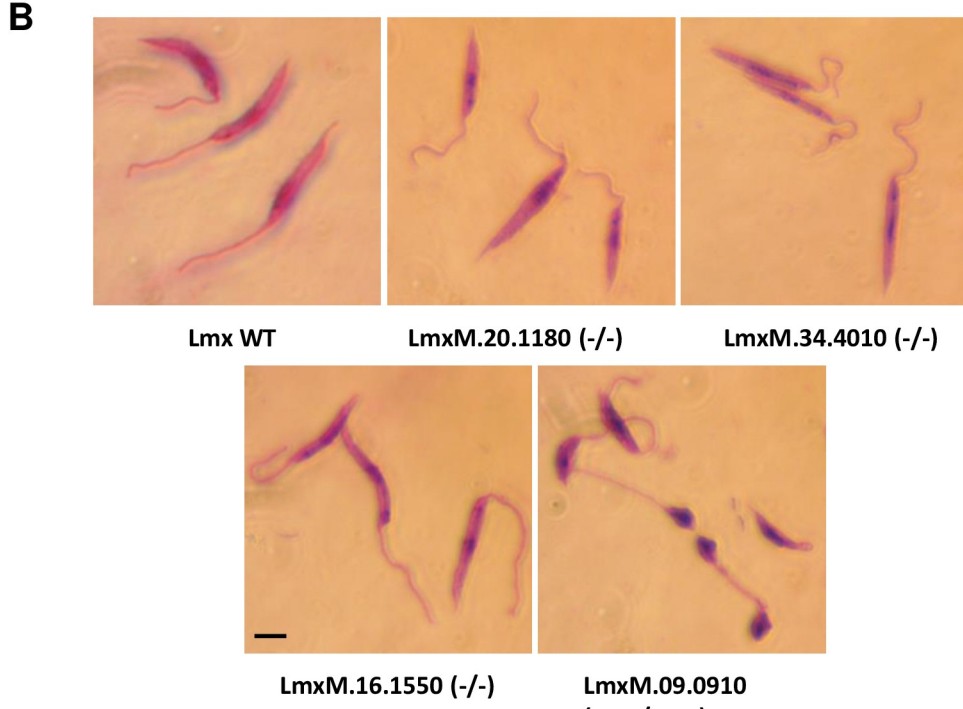

**Fig 6. Mobility defects were observed in *L. mexicana* null mutants generated in this study.** (A) A swimming assay was performed on a 7 cm VINYL Tubing with both ends in an up position as described in methods. The tube was first filled with 900 μl PBS, then 2 million *L. mexicana* promastigotes in 50 μL culture medium were gently loaded at the left end of the tube. The data shown in the table are the mean number (plus Standard Deviation, SD) of promastigotes detected in 3 μL solution taken from the right end of the tube after incubation for 1, 1.5, 2 and 3 hours. The formalin fixed *L. mexicana* promastigotes (Dead LmxM WT) was used as the negative control. The differences of promastigote number detected in the 3 μL solution between *L. mexicana* WT and the various mutant cell lines are statistically

significant (Student's t-test, * P<0.05; ** P< 0.005; *** P< 0.0005; **** P< 0.0001). (B) The Giemsa-stained *L. mexicana* promastigotes. The promastigote size and its flagellar length appear normal for the null mutants of LmxM.16.1550, LmxM.34.4010 and LmxM.20.1180. However, the promastigotes of *Calmodulin* CRISPR gene targeting clone (+—/—) are smaller than WT *L. mexicana* promastigotes though the normal flagellar lengths are still retained. Scale bar, 5 μm.

amplify other members of the gene family. This is important for example with the PKAC1 (LmxM.34.4010) and PKAC2 (LmxM.34.3960) gene family members that have similar but distinct sequences (see S2 and S3 Figs). Otherwise, even though the gene has been successfully targeted, the PCR analysis of that gene will show that the wild type gene has been retained. Consistent with the above observations, the generation of *L. mexicana PKAC1* and *PKAC2* null mutants with the T7 transient protocol has recently been reported using gene-specific PCR primers to verify the deletion mutants, and a similar phenotype as this study for the *PKAC* null mutants was observed [26].

The LeishGEM (*Leishmania* genetic modification project) is in progress (http://www. leishgem.org/) and represents an important resource to the *Leishmania* field. As this is a high throughput project to develop null mutants for all non-essential genes, it is necessary to use the high throughput T7 transient protocol [11]. The rRNA-P stable protocol may however be considered as a secondary approach when null gene mutants cannot be generated with the T7 transient protocol such as for example multi-copy genes or slow growing null mutants. Although it may be possible to generate slow growing null mutants using the rRNA-P stable protocol, it is nevertheless possible that the targeted gene may still be considered essential if compensating genome alterations take place to allow the null mutant to survive. Compensating genetic changes such as for example amplification of a different gene may be identified by whole genome sequencing and analysis of these genes can provide important insight into the function or biological pathways of the targeted genes.

CRISPR base editing has been recently adapted for use in *Leishmania* to introduce STOP codons in targeted genes [27]. Base editing differs from gene editing as it bypasses the DNA double strand break repair pathways, thus it requires no repair DNA donors and has significantly increased the gene inactivation efficiency through introduction of a premature STOP codon within coding sequence [27]. This advancement makes it feasible to construct the *Leishmania* loss of function CRISPR base editing libraries, though small sub-libraries may be required to overcome low transfection efficiency, and the extended culture time needed to generate the edited mutants. It is noteworthy that the plasmid vector used to perform base editing in *Leishmania* (pLdCH-hyBE4max) uses the pLdCH plasmid used in the rRNA-P stable protocol with the Cas9 gene replaced by the base editing Cas9 fusion gene (hyBE4max) [27]. Similar to the rRNA-P stable protocol described within, the base editing study reported that stable expression of gRNA and hyBE4max and prolonged culture were required for more efficient base editing [27].

In summary, for the large majority of *Leishmania* genes, the T7 transient CRISPR protocol is highly effective to generate null mutants, and this is particularly advantageous for high throughput gene targeting. The rRNA-P stable CRISPR protocol is highly effective to generate null mutants of genes with multiple copies and slow growing mutants. The more recent development of loss of function base editing will further expand the CRISPR technologies available for studying the *Leishmania* genome. Collectively, these complementary approaches have the potential to generate a wealth of knowledge about the function of the over 8000 genes in *Leishmania* genome for the development of novel treatments, vaccines and diagnostic tests.

## Materials and methods

### *Leishmania* strains and culture medium

*Leishmania mexicana* (MNYC/BZ/62/M379) and *L. donovani* 1S/Cl2D promastigotes were cultured at 27°C in M199 medium (pH 7.4) supplemented with 10% heat-inactivated fetal bovine serum, 40 mM HEPES (pH 7.4), 0.1 mM adenine, 5 mg l$^{-1}$ hemin, 1 mg l$^{-1}$ biotin, 1 mg l$^{-1}$ biopterine, 50 U mL$^{-1}$ penicillin, and 50 μg mL$^{-1}$ streptomycin. *Leishmania* promastigotes were passaged to fresh medium at a 40-fold dilution once a week.

### gRNA design and cloning

The gRNAs were designed with the aid of Eukaryotic Pathogen CRISPR guide RNA Design Tool (EuPaGDT) (http://grna.ctegd.uga.edu/) to avoid off target sites in the genome. A single gRNA guide coding sequence was ordered as standard oligos with 5′-TTGT and 5′-AAAC overhangs. All oligos and primers used in this study were ordered from Alphaadn (http://alphaadn.com/) or IDT (https://www.idtdna.com). The optimal guide length is 19 or 20 nt for SpCas9 gRNA and 21 nt for SaCas9 gRNA. After phosphorylation and annealing, the gRNA guide coding sequences were ligated into the *Bbs* I site of pLdCN or pLdSaCN CRISPR vector as described [19]. All oligos and primers used in this study are listed in S1 Table.

### Parasite transfection

$4 \times 10^7$ *Leishmania* promastigotes (middle log phase to early stationary phase) were harvested and washed once in 200 μL Tb-BSF buffer (90 mM Na$_2$HPO$_4$, 5 mM KCl, 0.15 mM CaCl$_2$, 50 mM HEPES, pH 7.3), and resuspended in 100 μL Tb-BSF buffer. Then, 2 to 5 μg CRISPR plasmid vectors in a volume < 20 μL were added and mixed. The transfection was performed in a 2-mm gap electroporation cuvette with the LONZA Nucleofector 2b Device (program U33). The transfected *Leishmania* cells were selected with 100 μg/mL G418 in the following day and selection takes 10 to 14 days to establish a resistant culture. Once the CRISPR plasmid vector transfected cell culture was established, those cells were then transfected with the antibiotic selection marker donor which was prepared by PCR and contains 25 bp homology arms to the Cas9 cleavage site. In the following day, 100 μg/mL Zeocin, or 30 μg/mL puromycin was added into the donor transfected cell culture. After 5 to 6 days incubation, the medium was replaced once with fresh medium before cloning into a 96 well plate. Note: The bleomycin resistance gene cassette conveys resistance to zeocin; the Puromycin resistance gene cassette conveys resistance to Puromycin.

### Cloning into 96-well plates

To prevent overgrowth of partially targeted *Leishmania* promastigotes in the culture flask, the double antibiotics (G418/ Zeocin or Puromycin) resistant promastigotes were cloned into a 96 well plate once enough cells were available for cloning, usually 10 to 21 days post donor transfection. Since Zeocin takes at least one to two weeks to completely kill the wildtype parasites, we waited about two weeks before cloning the bleomycin resistance gene donor transfected cells. After cloning into a 96 well plate, cell proliferation in each well of the plate would be monitored every two to three days. More attention was paid to the slow growing clones, which were then marked to facilitate following up. Since disruption of all the essential gene alleles would eventually result in cell death, cells in those wells could replicate slowly initially then stop growing once the gene product was diluted and degraded to the minimum level required for cell survival. If the gene product is required for optimal growth but non-essential, cells in those slow growing wells were most likely to be the gene null mutant clones. Fewer surviving

clones will be observed after cloning into a 96 well plate when targeting essential genes. Depending on the gRNA activity, the relative importance and abundancy of the essential gene product, 3–15 dying clones can usually be observed after the double antibiotic resistance transfectants are cloned into a 96 well plate. The dying clump size could vary depending on the relative importance and abundance of the essential gene product. More dying clones may be observed when large size dying clumps exist in a 96 well plate. It is also possible that more dying clones will be observed when an essential gene is targeted with a gRNA with higher activity.

If no dying clones was observed in the 96 well plate, 6–10 clones (especially the slow growing clones) were initially selected for PCR analysis. If necessary, more clones could be analyzed until a null mutant clone was detected. If many dying clones and empty wells were already observed in the 96 well plate, ideally all the remaining surviving clones or at least the 20 slower growing clones in the plate should be PCR analyzed to determine the gene essentiality.

### Genomic DNA preparation and PCR analysis

Parasite genomic DNA was extracted from *Leishmania* promastigotes with the minipreparation method as described [28]. The purity and quantity of those genomic DNA were assessed by Nanodrop spectrophotometer. Parasite genomic DNA could also be prepared with M-Fast PCR Genotyping kit (ZmTech Scientifique, Montreal). Briefly, 100 μL stationary phase *Leishmania* culture was harvested by centrifugation at 1,500 g for 5 min, and the cell pellet was resuspended in 15 μL reagent-A and incubated at 95°C for 30 min in a PCR apparatus. Once the tube cooled to room temperature, 3 μL reagent-B was added into the parasite tube and mixed well. Then, 0.5 μL lysate supernatant was added into 11.5 μL PCR mastermix for a total 12 μL reaction.

Primers were designed manually or using Primer3 (http://bioinfo.ut.ee/primer3-0.4.0/). Optimal primer length was 20 nucleotides with 60°C Tm. The *Taq* DNA polymerases used in this study include 2X DreamTaq Green PCR Master MIX, and 2X Platinum SuperFi PCR Master Mix (Thermo Fisher Scientific). The PCR program was set up according to manufacturer's instruction with variation in annealing temperature, extension time, and PCR cycles. The PCR products were separated in 1% to 1.5% agarose gel. If required, the specific PCR bands were extracted from the gel and sent to Genome Quebec Sequencing Center for sequencing confirmation.

### Growth curves and cell imaging

*Leishmania* promastigotes were seeded at the density of $1 \times 10^6$ promastigotes/mL in 4 mL M199 media in 25 cm$^2$ flasks and counted by a microscope daily for 8 days. A minimum of three biological replicates were evaluated. Movies and dead cell clumps were recorded with Nikon ECLIPSE TE200 Inverted Microscope and Nikon superhigh-performance 3 x zoom digital camera COOLPIX990. The Giemsa-stained *L. mexicana* promastigotes were examined with an OLYMPUS BH-2 light microscope and photographed with OMAX A35140U 14MP USB2.0 C-Mount Microscope Camera.

### Promastigote motility assay

A promastigote swimming assay was performed on a 7 cm vinyl Tubing (3/16 ID x 1/16 wall) with both ends in up position after turning the tube near 90 degrees at the ends in an Eppendorf tube rack (see Fig 5A). The tube was first filled with 900 μl PBS, then 2 million *L. mexicana* promastigotes in 50 μL culture medium were gently loaded at the left end of the tube. After incubation at room temperature for 1, 1.5, 2 and 3 hours, 3 μL solution was taken from

the right end of the tube for detecting with a microscope and quantitation of promastigotes reached the right end of the tube.

## Supporting information

**S1 Fig. Evidence for *Leishmania* essential genes included in Table 1 using the rRNA-P stable CRISPR protocol.** PCR shows the retention of the wild type gene for all surviving clones tested and at least one allele of the essential gene in the surviving clones (+/-) was successfully targeted and disrupted by CRISPR. Observation of the dying and dead promastigotes clumps of the essential gene null mutants' clones (-/-) in 96 well plates. In the rRNA-P stable CRISPR protocol, the gRNA and Cas9 are constantly expressed, the gRNA/Cas9 complex will continue to scan the genome until the last copy of the target gene is deleted or disrupted. If the gene is required for viability, once the remaining copy of the essential gene has been disrupted by CRISPR after cloning into a 96 well plate, null mutants in those wells would stop growing or multiply slowly to form clumps until the gene products are diluted and degraded to the minimum level required for survival. Depending on the relative importance, and initial abundance of the essential gene product in the cell and how soon the individual promastigote was cloned into a 96 well plate after the complete gene disruption, the dying (dead) cell clump size could vary from only a few cells to more than hundreds of cells, the earlier the complete gene disruption took place before cloning, the smaller the size of the dying cell clumps was expected. In this approach, by combining observation of the dying gene null mutant clones and detection of the WT gene band in all surviving clones, using rRNA-P CRISPR protocol, we provide evidence that 14 out of 22 genes listed in Table 1 are essential for promastigote viability. Those essential genes include ten *L. mexicana* kinase genes: LmxM.02.0290 (Mitogen-activated kinase kinase); LmxM.03.0780 (serine/threonine-protein kinase); LmxM.08.0530; LmxM.08_29.1330 (serine/threonine-protein kinase; Aurora kinase 2, AUK2); LmxM.17.0790 (polo-like protein kinase, PLK); LmxM.20.0960; LmxM.24.2010 (phosphatidylinositol 3-kinase, PI3K); LmxM.25.2340 (AGC essential kinase 1, AEK1); LmxM.30.2860 (Tousled-like kinase, TLK); LmxM.30.2960 (Repressor of differentiation kinase 2, RDK2) and the Calmodulin gene LmxM.09.0910; and three *L. donovani* genes LdBPK_111030 (hypothetical protein); LdBPK_260650 (Protein of unknown function (DUF2012)) and LdBPK_354780 (Hsp70 protein) (also see Figs 2 and 3).
(PDF)

**S2 Fig. Successful gene disruption of non-essential *L. mexicana* PKAC1 (LmxM.34.4010) and PKAC2 (LmxM.34.3960) with rRNA-P stable protocol.** (A) PKAC1 (LmxM.34.4010) and PKAC2 (LmxM.34.3960) genes are located in chromosome 34 and share large part of conserved sequences. (B) Strategy used by rRNA-P stable protocol to disrupt LmxM.34.3960 and LmxM.34.4010 genes. Use targeting PKAC1 as an example, a gRNA was designed to target the gene specific 5' end coding sequence of PKAC1 gene, which was then disrupted with the bleomycin resistance gene donor. (C) PCR analysis showing both PKAC1 gene alleles were successfully disrupted. (D) PCR analysis showing both PKAC2 gene alleles were successfully disrupted.
(PDF)

**S3 Fig. Successful gene disruption of *L. mexicana* CALP1 (LmxM.20.1180) with rRNA-P stable protocol.** (A) CALP1 (LmxM.20.1180) gene is located in chromosome 20 and shares the conserved sequence with the downstream CALP2 (LmxM.20.11185) gene. (B) Strategy used in this study to disrupt CALP1 gene. On the left panel, a gRNA was designed to target the specific 5' end coding sequence of CALP1 gene, which was then disrupted with the bleomycin

resistance gene donor. On the right panel, PCR analysis shows both CALP1 gene alleles were successfully disrupted.
(PDF)

**S1 Table. Oligonucleotides and primers used in this study.**
(DOCX)

**S1 Movie. *Leishmania mexicana* wildtype promastigotes in culture.**
(MOV)

**S2 Movie. LmxM.16.1550 (CMF6) null mutant promastigotes in culture.**
(MOV)

**S3 Movie. LmxM.34.4010 (PKAC1) null mutant promastigotes in culture.**
(MOV)

**S4 Movie. LmxM.09.0910 (*Calmodulin*) partial deletion mutant (+—/—) promastigotes in culture.**
(MOV)

**S5 Movie. LmxM.20.1180 (CALP1.1) null mutant promastigotes in culture.**
(MOV)

**S1 Raw images.**
(PDF)

## Author Contributions

**Conceptualization:** Wen-Wei Zhang.

**Data curation:** Wen-Wei Zhang.

**Formal analysis:** Wen-Wei Zhang.

**Funding acquisition:** Greg Matlashewski.

**Investigation:** Wen-Wei Zhang.

**Methodology:** Wen-Wei Zhang.

**Project administration:** Wen-Wei Zhang.

**Resources:** Greg Matlashewski.

**Validation:** Wen-Wei Zhang.

**Visualization:** Wen-Wei Zhang.

**Writing – original draft:** Wen-Wei Zhang, Greg Matlashewski.

**Writing – review & editing:** Wen-Wei Zhang, Greg Matlashewski.

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
