## [Decision Letter · Decision Letter 0]

28 Aug 2024

PONE-D-24-31151Determination of gene essentiality in Leishmania using CRISPRPLOS ONE

Dear Dr. Zhnag,

Thank you for submitting your manuscript to PLOS ONE. After careful consideration, we feel that it has merit but does not fully meet PLOS ONE’s publication criteria as it currently stands. Therefore, we invite you to submit a revised version of the manuscript that addresses the points raised during the review process.

At the heart of this work is a comparison of null-mutant generation between two different CRISPR methodologies, the transient T7 approach and the stable rRNA promoter-driven gRNA approach. It is interesting and somewhat informative that in several instances, the stable approach has successfully yielded null mutants, where the transient approach targeting the same gene was unsuccessful. However, this work would have been stronger if the authors performed a direct, side-by-side comparison of the two techniques instead of comparing their stable approach data to the published outcomes of transient-based methodologies. This point is discussed in detail by Reviewer 2.

Additional points:

In line with concerns raised by Reviewer 2, the authors conclude that “dying and dead cells were caused by the disruption of all wild type gene allele present in these clones”, however presumably due to the obvious difficulties inherent with this, they do not provide direct supportive evidence. 

The LmxM.20.1180 null mutant was described in the text as normal in mobility, yet the swimming assay data in Fig 5B indicate that it is more than twice as mobile as WT. This should be accounted for.

In Fig. 2D, please account for the additional fainter band running at 700 bp in the Cal (+--/---) lane.

Many minor issues with language throughout the manuscript that need to be edited.

In addition to these points, please also address all points raised by Reviewer 2.

We look forward to receiving your revised manuscript.

Kind regards,

Ben L. Kelly, Ph.D.

Academic Editor

PLOS ONE

Journal requirements: 1. When submitting your revision, we need you to address these additional requirements. Please ensure that your manuscript meets PLOS ONE's style requirements, including those for file naming. The PLOS ONE style templates can be found at https://journals.plos.org/plosone/s/file?id=wjVg/PLOSOne_formatting_sample_main_body.pdf and https://journals.plos.org/plosone/s/file?id=ba62/PLOSOne_formatting_sample_title_authors_affiliations.pdf. 2. Thank you for stating the following in the Acknowledgments Section of your manuscript: [This work was supported by the Canadian Institute of Health Research grant MOP125996 to G.M.]We note that you have provided funding information that is not currently declared in your Funding Statement. However, funding information should not appear in the Acknowledgments section or other areas of your manuscript. We will only publish funding information present in the Funding Statement section of the online submission form. Please remove any funding-related text from the manuscript and let us know how you would like to update your Funding Statement. Currently, your Funding Statement reads as follows:  [The author(s) received no specific funding for this work.] Please include your amended statements within your cover letter; we will change the online submission form on your behalf. 3. PLOS ONE now requires that authors provide the original uncropped and unadjusted images underlying all blot or gel results reported in a submission’s figures or Supporting Information files. This policy and the journal’s other requirements for blot/gel reporting and figure preparation are described in detail at https://journals.plos.org/plosone/s/figures#loc-blot-and-gel-reporting-requirements and https://journals.plos.org/plosone/s/figures#loc-preparing-figures-from-image-files. When you submit your revised manuscript, please ensure that your figures adhere fully to these guidelines and provide the original underlying images for all blot or gel data reported in your submission. See the following link for instructions on providing the original image data: https://journals.plos.org/plosone/s/figures#loc-original-images-for-blots-and-gels.   In your cover letter, please note whether your blot/gel image data are in Supporting Information or posted at a public data repository, provide the repository URL if relevant, and provide specific details as to which raw blot/gel images, if any, are not available. Email us at plosone@plos.org if you have any questions. 4. Please remove the supplementary figure in file "Figures V2.pdf".

Reviewers' comments:

Reviewer's Responses to Questions

**Comments to the Author**

1. Is the manuscript technically sound, and do the data support the conclusions?

Reviewer #1: Yes

Reviewer #2: Partly

2. Has the statistical analysis been performed appropriately and rigorously? 

Reviewer #1: N/A

Reviewer #2: No

3. Have the authors made all data underlying the findings in their manuscript fully available?

Reviewer #1: Yes

Reviewer #2: No

4. Is the manuscript presented in an intelligible fashion and written in standard English?

Reviewer #1: Yes

Reviewer #2: Yes

5. Review Comments to the Author

Reviewer #1: In this paper, Wen-Wei and Matlashewski investigate CRISPR as a tool for identifying essential genes in Leishmania. They compare the two commonly used CRISPR gene targeting methods in Leishmania: the stable expression of the gRNA and Cas9 using a plasmid containing a Leishmania ribosomal RNA gene promoter (rRNA-P stable protocol) and the T7 RNA polymerase-based transient gRNA expression system in promastigotes stably expressing Cas9 (T7 transient protocol). The authors set to determine whether the plasmid-based rRNA-P stable protocol could generate viable gene null mutants that were previously considered essential using the T7 transient system. Here, the rRNA-P stable protocol was used to target 22 Leishmania genes previously considered essential using the T7 transient protocol. Notably, the rRNA-P stable protocol generated surviving null mutants for 8 of the 22 genes and confirmed essentiality for the remaining 14 genes. The authors indicate that this study demonstrates the advantage of performing the rRNA stable protocol to confirm gene essentiality that can be performed alone or following high throughput gene targeting with the T7 transient protocol to identify candidate essential genes. The results from this study suggest that the rRNA stable protocol is more suitable for generating multi-copy gene null mutants and null mutants with reduced proliferation because the gRNA and Cas9 are stably expressed from the CRISPR plasmid and the transfectants are cloned. The authors conclude that for the majority of Leishmania genes, the T7 transient CRISPR protocol is highly effective in generating null mutants, and this is particularly advantageous for high throughput gene targeting. The rRNA-P stable CRISPR protocol is highly effective in generating null mutants of genes with multiple copies and a slow-growing phenotype that is particularly advantageous to confirm gene essentiality. Furthermore, the recent development of loss of function base editing will further expand the CRISPR technologies available for studying the Leishmania genome. Collectively, these complementary approaches have the potential to generate a wealth of knowledge about the function of the over 8000 genes in the Leishmania genome for the development of novel treatments, vaccines, and diagnostic tests.

This is a timely and very important paper that provides important information on one of the most used techniques: gene deletion. Moreover, we, the scientists in the field of leishmaniasis, struggle to be sure about the essentiality of the gene we delete from the parasite genome. To date, CHRISPR is the most used technology for gene deletion. This study provides an important and useful guide on this topic. The paper is well written and describes a carefully designed study. Hence, this paper should be accepted for publication as is.

Reviewer #2: This paper reports the results of gene deletion attempts with a specific CRISPR protocol (named rRNA-P “stable” protocol). 22 genes were targeted, which in previous studies that used a different CRISPR protocol (“transient” protocol) did not yield confirmed null mutants. The authors show that they successfully knocked out some of these genes, and for some they also failed to obtain null mutants.

The stated aim, broadly, is to determine whether the “stable” protocol could be used to “define genes as essential in Leishmania”. The authors conclude that “This study demonstrates the advantage of performing the rRNA stable protocol to confirm gene essentiality […].” (line 128) This conclusion is flawed. Fundamentally, failure to generate a viable knock-out is never sufficient to prove that said gene is “essential”. Jones et al. 2018 (PMID: 29384366) have provided the most comprehensive analysis to date of reverse genetically engineered knockout mutants in Leishmania. They rehearsed in detail how a combination of different genetic approaches can increase the confidence with which a claim of “essentiality” can be made. That landmark study was not cited, but these considerations are highly relevant.

The attempts to compare the “stable” and the “transient” methods are flawed too. Using the two methods in parallel to target the same genes and compare results would allow for a comparison. Instead, results were taken from studies with different goals (generating large numbers of KOs, prioritising throughput) and compared here against the goal of isolating null mutants for a small handful of genes. This is a comparison of two very different experimental workflows from which few conclusions can be drawn about the relative power of these methods to achieve gene knockouts. The key difference is clonal vs. population analysis, rather than “stable” vs. “transient” CRISPR methods.

A clarification is required about the “stable strategy”: Cells transfected with a plasmid that contains both Cas9 and the gRNA were selected for a period of time. How much time – several days, a week? During this time, gRNA-complexed Cas9 will presumably cut the target locus. Some alleles will be mutated, many will be repaired though homologous recombination with the other allele, but can be cut again. What is the status of the target locus after stable expression of the Cas9-gRNA plasmid, but BEFORE introduction of the donor cassettes? The worry is that non-lethal mutations in the target locus, or reduced gene dosage, could promote adaptation to the loss of gene function. This would facilitate a subsequent full deletion. This is another marked difference in protocol, which makes comparisons between the methods, as presented, additionally challenging.

There is no doubt that different CRISPR protocols have different strengths and weaknesses and they should be used in complimentary ways. Particularly in situations of multi-copy gene families “stable” strategies, such as the one described here, will be invaluable. This is well elaborated in the discussion.

For the data itself, the conclusions form the successful gene deletions with the “stable” method are sound. The conclusions from the unsuccessful attempts are not all that well supported; important controls are missing.

Figure 1 shows that two genes that did not yield KOs in Ref 21, which used the “transient” method were successfully replaced by the “stable” method. The data showing loss of the WT band is sound.

Figure 2B. The death of clonal lines is taken as evidence of successful gene deletion. This is perhaps a reasonable assumption, but not proven. Death of the cells does not in itself “indicate all the calmodulin genes [..] had been deleted or disrupted” as stated in line 224. Observing that cells subjected to a CRISPR protocols ended up dead is neither proof of gene deletion (the DNA was not analyzed), nor of gene “essentiality” (ref. Jones et al. 2018). The level of evidence is the same whether this death occurs quickly (in the “transient” protocol) or slowly, as shown here with the “stable” protocol.

Figure 2D. The evidence that this clone has both WT and modified calmodulin loci is sound. The conclusion that the genotype is +-- / --- does not follow from these data. The PCR is not quantitative, the ratio of WT vs. modified copies cannot be estimated. Even the possibility that the locus has been amplified (perhaps with a point mutation in the sgRNA targeting site) cannot be excluded. Minimally, the statement “one of the calmodulin genes remains intact” (line 228) should be changed to “at least one…” (as done correctly in line 292). In short, whole genome sequencing of this clone should be done to clarify what the locus looks like in this clone.

It is also unclear how stable this genotype is, given the dynamic nature of the Leishmania genome. The quality of the microscopy images in Figs 2C-D is poor, but these round cells look markedly different from the flagellated ones shown in Figure 5B. Additionally, the growth curve in Fig 4 shows they grow at the same rate as WT. This would be surprising given the massive effect on morphology. This cell line could be valuable for studying the functions of calmodulin (outside of the scope of this paper). As it is, the presented data lacks sufficient information about the modified gene locus to conclude much, except that this clone is not a null mutant. This is largely the same conclusion that was drawn from the incomplete deletion of this gene by the “transient” method. Whether this failure to remove the gene completely is technical or biological (“essential” gene) cannot be proven by either method, without additional corroborating evidence.

Figure 3. Knockout of genes on supernumerary chromosomes.

Gene on chromosome 31 – the PCR result looks convincing.

CMF6. All alleles of the gene were lost in some clones following the “stable” protocol. The gene was removed in a minority of the tested clones. This conclusion is supported by the PCR result. The statement that presence of the gene in many clones indicates pressure to retain it is not strongly supported by the evidence. An alternative explanation is that the failure is technical: there could be sequence variation at this locus (perhaps more likely in a trisomic chromosome?) preventing recognition by the gRNA. Or the repair events that happened while cells had the Cas9 plasmid (before addition of the donor DNA) led to small sequence changes that made some alleles refractory to subsequent deletion. The sequence of the remaining allele should be determined to exclude these possibilities. The observation that the null mutants grew slower, is the most convincing evidence supporting the idea that the gene is required for normal growth.

Referring to Line 248 (also 265 ff): The idea that cloning the cells after transfection helps to isolate the null mutant from such mixed populations cases is correct. This is not a new idea and it is equally applicable to slow (“stable”) or fast acting (“transient”) gene deletion protocols. Beneke and Gluenz already published a detailed protocol for clone selection after transfection with the “transient” protocol in 2019 (PMID: 30980304).

Figure 5. The swimming assay in A lacks a negative control that cannot swim. An immobilized WT or a paralyzed or aflagellate mutant could be used to show how passive movement impacts on the recovery of cells from the other side of the tube. These data in themselves may indicate differences in motility, but are not compelling as presented. The results for PKAC1 are however consistent with the data by Fochler et al. Biorxiv 2023 that showed (i) that it was possible to isolate a PKAC1 null mutant also with the “transient” method, and (ii) this mutant was defective in flagellar beating.

Other comments:

The study design is based on the statement that the selected genes had been targeted unsuccessfully for knockout by CRISPR in one of three previous studies (refs 19-21 in the paper). The claim that these were “considered to be essential” is incorrect and should be modified, especially with regards to Ref 19, which did not comment on “essentiality” of any genes.

Consideration should be given to the biological functions of the targeted genes. This information should be added to Table 1.

line 229, “Additional PCR primers (not shown) had been used to verify the gene targeting outcome shown”. Please clarify what this means and include the relevant data or remove the statement.

Line 244, “triploid” means an extra set of chromosomes. Here, the correct term to describe the extra copy of chromosome 16 is “trisomic”.

Figure 4 please plot growth curves of exponentially growing cells on a semi-logarithmic graph (Y-axis on a log scale). Were doubling times calculated and which differences were significant compared to WT?

line 304 “its flagellum length appeared normal” – how was this quantified?

Line 439 – “dying crump” - do you mean clump?

Figure 5. The table in A lacks information. Do the numbers really represent numbers of Leishmania in 3µl (i.e. there were samples that contained 1 parasite)? How many times was each mutant measured? What statistical tests were done to support the statement that some mutants were slower?

6. PLOS authors have the option to publish the peer review history of their article (what does this mean?). If published, this will include your full peer review and any attached files.

Reviewer #1: **Yes: **Dan Zilberstein

Reviewer #2: No

---

## [Author Response · Author response to Decision Letter 0]

4 Oct 2024

Responses to Editor Dr. Kelly

Editor Comment. At the heart of this work is a comparison of null-mutant generation between two different CRISPR methodologies, the transient T7 approach and the stable rRNA promoter-driven gRNA approach. It is interesting and somewhat informative that in several instances, the stable approach has successfully yielded null mutants, where the transient approach targeting the same gene was unsuccessful. However, this work would have been stronger if the authors performed a direct, side-by-side comparison of the two techniques instead of comparing their stable approach data to the published outcomes of transient-based methodologies. This point is discussed in detail by Reviewer 2.

Response: We agree it is better to perform a direct, side-by-side comparison between the T7 transient protocol and the rRNA-P stable protocol. Ideally, the direct side by side comparison should be carried out by a third-party lab to avoid bias since our lab developed the rRNA-P stable protocol. Nevertheless, the T7 transient protocol studies (20-22) were performed from well-established Leishmania research labs with a minimum three or more attempts for each of the genes outlined in Table 1. Consequently, we do not believe the outcome would be different if these analyses were repeated in our lab. See also further explanations in responses to Reviewer 2 below.

Editor Comment: In line with concerns raised by Reviewer 2, the authors conclude that “dying and dead cells were caused by the disruption of all wild type gene allele present in these clones”, however presumably due to the obvious difficulties inherent with this, they do not provide direct supportive evidence. 

Response: We understand this concern. We have attempted to extract the genomic DNA from those dying and dead Leishmania cells in the wells of 96 well plates. However, due to the extreme low yield and poor quality (degraded DNA), we were not able to obtain PCR products. We reasonably believe the disruption of all wild type essential gene alleles in these clones is the most likely cause of the cell dying and death. To better explain and illustrate how gene essentiality is normally determined with the rRNA-P stable protocol, we have now rewritten the section of essential genes (line 170-232) using the AGC essential kinase 1 gene (LmxM.25.2340, AEK1) as a typical example (see new Fig. 2).

See also response below to reviewer 2 that further explains the interpretation of dead and dying clones.

Different from the T7 transient protocol, the gRNA and Cas9 are constantly expressed in the stable CRISPR protocol. Once the CRISPR plasmid pLdCN (pLdsaCN) was transfected into Leishmania promastigotes, the gRNA/Cas9 complex would continue scanning the genome until all the WT alleles were targeted and disrupted. If the gene is required for viability and once the remaining copy of the essential gene has been disrupted by CRISPR, cells in those wells would stop growing or multiply slowly to form clumps until the gene products (mRNA and protein) are diluted and degraded to the minimum level required for survival. Depending on the relative importance, initial abundance, and stability of the gene product in the cell and the cloning time (stage) post the complete gene disruption for the individual clone, the dying (dead) cell clumps could contain only a few cells to more than hundreds of cells (new Fig. 2A and see below). (Note: if the cell dying was caused by the failed double strand DNA break repair, the promastigote would stop proliferating right after cloning and would not be able to form clumps containing more than ten to hundreds of cells; if the cell dying was caused by a spontaneous rare lethal mutation in the genome (unlikely), much fewer dying clones (maybe only one clone if it did happen) would be observed in a 96 well plate, instead of more than three dying clones regularly observed for disruption of an essential gene). PCR analysis will show the WT gene band persists in all surviving clones and at least one allele of the essential gene was successfully targeted and disrupted by CRISPR (new Fig. 2B). In this manner by combining observation of death of gene null mutant clones and detection of the WT gene band and the gene targeting band in all surviving clones, using rRNA-P stable protocol, we were able to confirm with confidence that many of those genes which T7 system was not able to generate alive null mutants (14 out of 22, see table 1, new Fig 2, new Fig 3 and S1 Fig ) are truly essential for Leishmania viability. 

Editor Comment: The LmxM.20.1180 null mutant was described in the text as normal in mobility, yet the swimming assay data in Fig 5B indicate that it is more than twice as mobile as WT. This should be accounted for.

Response: We agree that a more precise description should be added to the text regarding LmxM.20.1180 null mutant. We have now revised the sentence (line 297-299) as below:

In comparison, the LmxM.20.1180 (CALP1.1) null mutants are normal in size but appeared to be able to proliferate and swim slightly faster than the wild type cells (S5 movie and Figs 5 and 6).

 Editor Comment: In Fig. 2D, please account for the additional fainter band running at 700 bp in the Cal (+--/---) lane.

Response: For simplicity we did not include the detailed PCR analysis data in the old Figure 2 in the version submitted. We have now rewritten the text regarding the Calmodulin gene targeting and included the detailed PCR analysis data on the Calmodulin (+--/---) clone with additional PCR primers in the new Figure (now Figure 3) (line197-232). As you can see in Fig. 3, PCR analysis revealed that CRISPR gene targeting had deleted both alleles of LmxM 09.0930, disrupted both alleles of LmxM 09.0910 and one allele of LmxM 09.0920, and only one wild type LmxM 09.0920 allele remained in this slowest growing clone (Fig. 3B&C). It is interesting to note: a smaller than the expected size (R+10R1) band was detected in this Cal (+--/---) clone, indicating a recombination deletion had occurred in one of the two disrupted LmxM 09.0910 alleles, which also explains the additional fainter band running at 700 bp detected in the PCR with (R+L) primer pair. 

Editor Comment: Many minor issues with language throughout the manuscript that need to be edited.

Response: We have carefully gone through the manuscript and made corrections if necessary.

Editor Comment: In addition to these points, please also address all points raised by Reviewer 2.

Response: We have responded to all concerns raised by Reviewer 2. 

Reviewer 1

Comment: This is a timely and very important paper that provides important information on one of the most used techniques: gene deletion. Moreover, we, the scientists in the field of leishmaniasis, struggle to be sure about the essentiality of the gene we delete from the parasite genome. To date, CHRISPR is the most used technology for gene deletion. This study provides an important and useful guide on this topic. The paper is well written and describes a carefully designed study. Hence, this paper should be accepted for publication as is.

Response: We thank Reviewer 1 for his understanding and support of this study.

Reviewer 2

Comment: This paper reports the results of gene deletion attempts with a specific CRISPR protocol (named rRNA-P “stable” protocol). 22 genes were targeted, which in previous studies that used a different CRISPR protocol (“transient” protocol) did not yield confirmed null mutants. The authors show that they successfully knocked out some of these genes, and for some they also failed to obtain null mutants.

Response: We agree.

Comment: The stated aim, broadly, is to determine whether the “stable” protocol could be used to “define genes as essential in Leishmania”. The authors conclude that “This study demonstrates the advantage of performing the rRNA stable protocol to confirm gene essentiality […].” (line 128) This conclusion is flawed. Fundamentally, failure to generate a viable knock-out is never sufficient to prove that said gene is “essential”. Jones et al. 2018 (PMID: 29384366) have provided the most comprehensive analysis to date of reverse genetically engineered knockout mutants in Leishmania. They rehearsed in detail how a combination of different genetic approaches can increase the confidence with which a claim of “essentiality” can be made. That landmark study was not cited, but these considerations are highly relevant.

Response: The literature review article by Jones et al. 2018 (PMID: 29384366) proposes a 5-star method for determining gene essentiality which we generally agree with and have now included this reference in the revised paper (ref 2). In practice however, it is not practical to meet the 4–5-star stringent criteria proposed by Jones et al using forced plasmid shuffling and DiCre gene deletion methods which are labor intensive and time consuming, for which there are only a few examples in the literature for Leishmania. Actually, the rRNA-P stable CRISPR is similar to the DiCre inducible gene deletion method as both methods rely on observation of cell death after complete disruption or deletion of the essential genes. 

Because of its simplicity and effectiveness, CRISPR has revolutionized gene targeting methods in almost all living organisms. We believe it is also time to embrace the new CRISPR technology for determining gene essentiality in Leishmania. For example, Dr. J.C. Mottram group, the authors of the above review article, has set up a new diagnosis standard for essential genes with the T7 CRISPR transient protocol; that is a gene can be considered essential if the alive null mutant could not be generated after minimum three attempts of the T7 transient method (see nature communication (reference 21) “If a gene deletion mutant was unable to be identified after three transfection attempts, it was classed as ‘required’ with a 1-star quality classification. Forty-three protein kinases and AMPKγ fell into this category”) [21]. As verified by our stable CRISPR protocol in this study, three attempts’ method did correctly predict the gene essentiality for 10 of the 12 kinase genes which the T7 transient protocol failed to isolate the alive null mutants after three attempts in reference 21 study. If the PKAC1 and PKAC2 are excluded because failure to generate those null mutants were likely due to the primer design error in the references 20 and 21 studies, the predication accuracy is 100% (10/10). However, verified by this study, this three attempts’ method has only 43% (3/7) accurate rate in predicting the gene essentiality for 7 L. donovani genes from reference 22 study. Therefore, we believe it is important to understand the advantages and weaknesses of the T7 transient method and the rRNA-P stable method in targeting Leishmania genes and determining their essentiality. As demonstrated in this study, (the exact statement in line 128) “These results suggest that the rRNA-P stable protocol is a useful complement to the T7 transient system for investigating gene essentiality and may be more suitable for targeting multicopy genes.” 

This mentioned reference (below) has now been cited as reference 2

Jones, N. G., Catta-Preta, C. M. C., Lima, A. P. C. A. & Mottram, J. C. Genetically validated drug targets in Leishmania: current knowledge and future prospects. ACS Infect. Dis. 4, 467–477 (2018).

Consistent with the Jones review article (ref 2) we have changed the title of the paper to: “Evidence for gene essentiality in Leishmania using CRISPR”

Regardless of how essential genes are classified in Leishmania, this paper highlights one important consideration. As described, the rRNA stable protocol should be considered to generate the null mutant when it is not possible to generate a null mutant with the T7 transient protocol. The ability to generate null mutants is essential for studying differentiation, virulence, pathogenesis and basic biology of the parasite. This is now highlighted in the discussion of the revised version on lines 338-342.

Comment: The attempts to compare the “stable” and the “transient” methods are flawed too. Using the two methods in parallel to target the same genes and compare results would allow for a comparison. Instead, results were taken from studies with different goals (generating large numbers of KOs, prioritising throughput) and compared here against the goal of isolating null mutants for a small handful of genes. This is a comparison of two very different experimental workflows from which few conclusions can be drawn about the relative power of these methods to achieve gene knockouts. The key difference is clonal vs. population analysis, rather than “stable” vs. “transient” CRISPR methods.

Response: As our response to editor, to avoid preference, ideally, comparison between the stable and transient CRISPR methods should be performed in parallel to target the same batch of genes by a third-party laboratory. 

We understand generating hundreds of Leishmania gene Knockouts with the T7 transient method were challenging in the previous three studies (see reference 20,21and 22). However, it is not true that less effort was put to isolate the individual gene null mutants in those three high throughput studies. As mentioned above, a minimum of three transfection attempts of T7 transient protocol were performed in order to class a gene as essential according to the three attempts diagnosis standard (see reference 21). In fact, 5 transfections were attempted for the non-essential L. donovani genes LdBPK_100590 and LdBPK_230540 but the T7 transient method somehow failed to generate the null mutants. Furthermore, for unknown reasons, the T7 transient method was still not able to generate the chromosome alleles deletion mutant for LdBPK_100590 gene even after an episomal copy of the gene was provided to the parasite (see Fig. 2 in reference 22). 

For LmxM.34.4010 gene (PKAC1), likely because both the authors of reference 20 and 21 (two of the best Leishmania research labs in the world) did not realize that PKAC1 and PKAC2 (LmxM.34.3960) possess a large identical sequence at 3’ end (see S2 Fig), the primers used to verify the gene deletion in reference 20 and 21 were from the sequence shared by both PKAC1 and PKAC2. Therefore, detection of the WT gene band after targeting with the T7 system could be due to the non-specific primers used for PCR analysis for those genes (see reference 21). Regrettably, detection of the gene band shared by PKAC1 and PKAC2 after targeting PKAC1 gene with the T7 transient method has been used as an example how an essential gene was determined by PCR analysis (see Fig.2b in reference 21). A similar mistake was made in targeting LmxM.20.1180 gene in reference 20 study (see S3 Fig). 

It could be true that if cloning was applied, the null mutants could have been isolated by the T7 transient protocol for some of the five non-essential genes identified by the rRNA-P stable system in this study. However, because two antibiotic repair donors were typically used in the T7 transient method, indeed, null mutants for many non-essential genes could be obtained without cloning if these non-essential genes are present in diploid chromosomes and their deletions have no inhibition effect on the parasite growth such as most of the flagellar protein genes, this has given the impression to Leishmania research community that no cell cloning is needed when using T7 transient CRISPR method other than repeating transfections. Therefore, our advice is that one can continue using T7 transient method without cloning for the first attempt. However, if the null mutant cannot be obtained after the first try, it might be more efficient to clone the transfectants following the second transfection attempt rather than repeating the transfection many times without cloning. If after cloning, a null mutant still cannot be isolated by the T7 transient method, then it is time to consider the stable CRISPR expression approach.

It is important to note that this study was initiated by our curiosity why the deletion mutants for four of the Leishmania flagellar protein genes could not be generated by the T7 transient method. After blast search and sequence analysis, we realized that fai

---

## [Decision Letter · Decision Letter 1]

22 Oct 2024

PONE-D-24-31151R1Evidence for gene essentiality in Leishmania using CRISPRPLOS ONE

Dear Dr. Zhnag,

Thank you for submitting your manuscript to PLOS ONE. After careful consideration, we feel that it has merit but does not fully meet PLOS ONE’s publication criteria as it currently stands. Therefore, we invite you to submit a revised version of the manuscript that addresses the points raised during the review process.

We look forward to receiving your revised manuscript.

Kind regards,

Ben L. Kelly, Ph.D.

Academic Editor

PLOS ONE

**Journal Requirements:**

Reviewers' comments:

Reviewer's Responses to Questions

**Comments to the Author**

1. If the authors have adequately addressed your comments raised in a previous round of review and you feel that this manuscript is now acceptable for publication, you may indicate that here to bypass the “Comments to the Author” section, enter your conflict of interest statement in the “Confidential to Editor” section, and submit your "Accept" recommendation.

Reviewer #2: All comments have been addressed

2. Is the manuscript technically sound, and do the data support the conclusions?

Reviewer #2: Yes

3. Has the statistical analysis been performed appropriately and rigorously? 

Reviewer #2: Yes

4. Have the authors made all data underlying the findings in their manuscript fully available?

Reviewer #2: Yes

5. Is the manuscript presented in an intelligible fashion and written in standard English?

Reviewer #2: Yes

6. Review Comments to the Author

**Reviewer #2:** The revised manuscript addresses the main issues raised in the review of the initial submission and provides additional control experiments, which were needed to support the conclusions.

The detailed responses to the reviewers’ comments were particularly helpful in clarifying some interpretations of the data and the revised figures are easier to follow.

The claims around “proving” gene essentiality are better justified – the “stable” CRISPR protocol is (mostly) presented as a valuable tool to support claims of essentiality, rather than a tool to determine essentiality.

Importantly, there is good evidence in this paper that the “stable” CRISPR protocol is suitable to determine the KO phenotypes of multicopy genes. In my opinion this is the greatest strength of this method, which no doubt is a very useful tool for reverse genetics studies on Leishmania.

Most of the conclusions are supported by the data:

Figure 1 convincing evidence for successful KO of Ld100590 and Ld230540.

Figures 2 and 3 provide data that are consistent with essential functions for LmxM.25.2340 and calmodulin.

Figure 4. Supports the conclusion that Ld310120, Ld312380 and LmxM.16.1550 were knocked out and provides evidence that the stable protocol can be used to target genes on chromosomes with >2 copies successfully.

Figure 5. Supports the conclusion that some mutants grow more slowly as promastigotes. (The cell density [cells/ml] should be plotted on a log scale because a semi-logarithmic plot allows for the direct comparison of the growth rates during the exponential growth phase.)

Figure 6 supports the conclusion that some mutants are impaired in their motility.

There are still a few sections where clarification would be helpful (some of this information was provided in the extensive response to reviewers – here my suggestions where it would be helpful to add explanations to the manuscript text):

(1) Interpretation of dying cell clumps.

Figure 2. Microscopic observation shows cell clumps indicative of dying cells. The images support the conclusion that these clonal cell lines are not viable. 2B shows PCR products indicating the presence of the wild type LmxM.25.2340 gene as well as the modified gene locus. The interpretation is unclear. If the wild type gene is still detected, these are not KOs. The authors should clarify their interpretation of the PCR results and amend the figure legend as needed. Specifically:

- Did all clones die and form clumps as shown in 2A? Or did some wells still contain live cells that looked different from the clumps?

- Was the PCR in 2B done from wells with clumps (at what time point), or from wells with live cells?

- Do they conclude that one allele is insufficient for survival? Or did they conclude that cells that manage to survive until the final allele is removed. (But for technical reasons DNA can only be analyzed from cells that have not yet reached that final stage).?

There are several other sections where the assumption is made that cells were null mutants but the actual status of the gene locus could/was not determined:

line 45 (and 138-140) – “by directly observing gene null mutant promastigotes dying in culture”. The observation is: dying promastigotes. When these genes are targeted with the transient protocol, dying promastigotes are also observed. Whether or not the targeted gene is absent in these promastigotes is not known, in either case. All that can be concluded (in both protocols) is that failure to recover live cells is consistent with an essential function of the targeted gene (but not definitive proof).

Figure S1. line 647-648. “Morphology of clumping dying clones (-/-) once the remaining copy of the gene has been disrupted by CRISPR.” The PCR shows that the WT band was present in all samples. No evidence is presented to show that the remaining copy of the gene has been disrupted. It is possible that this is what happens, likely even in many cases, but the data does not establish a direct link between the loss of the gene and the death of the cells. If the authors’ interpretation was correct, there should be a decrease of the WT PCR product over time. There is no evidence that the WT band decreases with time.

The time point at which the DNA was tested should be stated.

Figure S1. line 647-648. These data do not provide evidence that the gene is essential. Other explanations for the death of the cells cannot be excluded.

For all of these, the only claim that can be made is that the observed decay of the cells is consistent with an essential function of the gene.

Note, Figure S1 legend. Lines 650-653 are a duplication of lines 224-227 in the main text.

(2) Ability to delete protein kinase A

Table 1. Null mutants for LmxM.34.3960 and LmxM.34.4010 (Protein kinase A catalytic

subunit isoforms) were not confirmed in the cited reference but they were subsequently reported in Fochler et al., Biorxiv 2023 (for completeness this could be added to footnote 6). This successful KO also used the transient protocol; it is likely it was able to confirm the KO because gene-specific primers were used, unlike the reference cited in the manuscript.

(3) Claims about calmodulin

Figure 3B, C, line 253-254. I do not follow the argument that “only one wild type LmxM.09.0920 allele remained”. There are 3 PCR bands in the mutant – it would be helpful to label them in the illustration (WT, modified locus, ...?).

How was allele copy number determined?

Line 250 “data not shown” – please show data or remove statement.

Line 257 “Calmodulin is only one of the 98 Leishmania flagellar protein genes” required for viability. It isn't clear if this means “only one of many” or “the only one”.

7. PLOS authors have the option to publish the peer review history of their article (what does this mean?). If published, this will include your full peer review and any attached files.

Reviewer #2: No

---

## [Author Response · Author response to Decision Letter 1]

2 Dec 2024

Response to New additional Comments by Reviewer2

Reviewer Comment: Figure 5. Supports the conclusion that some mutants grow more slowly as promastigotes. (The cell density [cells/ml] should be plotted on a log scale because a semi-logarithmic plot allows for the direct comparison of the growth rates during the exponential growth phase.)

Response: As we explained in the previous Response to reviewers, depending on the researcher’s preference, we understand that Leishmania growth curves can also be plotted on a semi-log plot. However, like most Leishmania research labs, we prefer to use the regular plot (the actual cell density; the number of promastigotes /ml) to plot the growth curve. As shown in the supplementary PowerPoint slide (for review purpose only) of comparison between the regular plot and the semi-log plot, it clearly shows that the regular plot can more precisely display the Leishmania growth rate and is easier to distinguish the differences between cell lines.

Reviewer Comment: There are still a few sections where clarification would be helpful (some of this information was provided in the extensive response to reviewers – here my suggestions where it would be helpful to add explanations to the manuscript text):

(1) Interpretation of dying cell clumps.

Figure 2. Microscopic observation shows cell clumps indicative of dying cells. The images support the conclusion that these clonal cell lines are not viable. 2B shows PCR products indicating the presence of the wild type LmxM.25.2340 gene as well as the modified gene locus. The interpretation is unclear. If the wild type gene is still detected, these are not KOs. The authors should clarify their interpretation of the PCR results and amend the figure legend as needed. Specifically:

- Did all clones die and form clumps as shown in 2A? Or did some wells still contain live cells that looked different from the clumps?

- Was the PCR in 2B done from wells with clumps (at what time point), or from wells with live cells?

- Do they conclude that one allele is insufficient for survival? Or did they conclude that cells that manage to survive until the final allele is removed. (But for technical reasons DNA can only be analyzed from cells that have not yet reached that final stage).?

There are several other sections where the assumption is made that cells were null mutants but the actual status of the gene locus could/was not determined:

Figure S1. line 647-648. These data do not provide evidence that the gene is essential. Other explanations for the death of the cells cannot be excluded.

For all of these, the only claim that can be made is that the observed decay of the cells is consistent with an essential function of the gene.

General response to the above reviewer comments (See specific responses below): We believe we have explained and answered the similar concerns in the previous Response to reviewers. We will explain here in detail once more, which will also answer some of the specific questions. Different from the T7 transient protocol, the gRNA and Cas9 are constantly expressed in the stable CRISPR protocol. Once the CRISPR plasmid pLdCN (pLdsaCN) was transfected into Leishmania promastigotes, followed by the antibiotic donor transfection, the gRNA/Cas9 complex would continue scanning the genome until all the WT alleles were targeted and disrupted. If the gene to be targeted is non-essential for viability, the cloned cells would continue to proliferate. Depending on the gene and/or gRNA, PCR analysis of those alive clones will show that either one gene allele or both gene alleles have been disrupted with the antibiotic selection marker, and the smaller WT gene band will not be detected in many of those clones (Fig. 1B). However, if the gene is required for viability and once the remaining copy of the essential gene has been disrupted by CRISPR, cells in those wells would stop growing or multiply slowly to form clumps until the gene products (mRNA and protein) are diluted and degraded to the minimum level required for survival. Depending on the relative importance, initial abundance, and stability of the gene product in the cell and the cloning time (stage) post the complete gene disruption for the individual clone, the dying (dead) cell clumps could contain only a few cells to more than hundreds of cells (Fig. 2A and S1 Fig). Three to ten dying null mutant clones will usually be observed in a 96 well plate when targeting an essential gene. PCR analysis of the remaining surviving clones in the 96 well plate will show the WT gene band persists in all surviving clones and at least one allele of the essential gene was successfully targeted and disrupted by CRISPR (Fig. 2B). In this manner, by combining observation of the death of gene null mutant clones and detection of the WT gene band and the gene targeting band in all surviving clones, using rRNA-P stable CRISPR protocol, we were able to confirm that many of those genes for which the T7 transient protocol was not able to generate alive null mutants (14 out of 22, see table 1, Fig 2, Fig 3 and S1 Fig) are truly essential for Leishmania viability. 

As suggested by the reviewer, we have revised and added the following sentences into the manuscript text to better explain how the gene essentiality is normally determined with the rRNA-P stable CRISPR protocol: 

Line208-210: “To illustrate how gene essentiality is normally determined with the rRNA-P stable protocol, we use targeting AGC essential kinase 1 gene (LmxM.25.2340, AEK1) as an example (Fig 2).”

Line228-233: “In this manner, by combining observation of the death of gene null mutant clones and detection of the WT gene band and the gene targeting band in all surviving clones, using the rRNA-P stable CRISPR protocol, we were able to confirm that many of those genes for which the T7 transient protocol was not able to generate alive null mutants (14 out of 22; see Table 1, Fig. 2, Fig. 3, and S1 Fig.) are truly essential for Leishmania viability.” 

Comment: - Did all clones die and form clumps as shown in 2A? Or did some wells still contain live cells that looked different from the clumps?

Response: No, not all clones in the 96 well plate form clumps and eventually die as shown in Fig 2A; most wells in the 96 well plate still contain live cells that looked different from the clumps. These live cell wells are the wells that were used to generate the PCR results in Fig. 2B. 

Although the gRNA and Cas9 are constantly expressed in the stable rRNA-P protocol, CRISPR gene targeting is still not very efficient in Leishmania. As described above, depending on the gene and the gRNA activity, 3 to 10 wells with dying null mutant clones(-/-) will usually be observed in a 96 well plate after the double antibiotic (one from pLdCN CRISPR vector, other from the antibiotic resistance repair donor) resistance promastigotes are cloned when targeting an essential Leishmania gene (Fig 2A, Fig 3D and S1 Fig). The live promastigotes (+/-) in the remaining wells will continue proliferation with various growth rate as CRISPR is still targeting the remaining essential gene allele in some of the cells to yield sufficient cells for subsequent PCR analysis. PCR analysis of those surviving clones in the 96 well plate will show the WT gene band persists and at least one allele of the essential gene was successfully disrupted by CRISPR (Fig. 2B and S1 Fig).

Comment: Was the PCR in 2B done from wells with clumps (at what time point), or from wells with live cells?

Response: The PCR in Fig. 2B (in all PCR analysis in this manuscript) was performed from wells with live cells, which is clearly stated in the text (line 226-228): “PCR analysis of all the surviving clones in the 96 well plate will show the WT gene band persists and at least one allele of the essential gene was successfully targeted and disrupted by CRISPR (Fig 2B, S1 Fig and below)”.

Comment: Do they conclude that one allele is insufficient for survival? Or did they conclude that cells that manage to survive until the final allele is removed. (But for technical reasons DNA can only be analyzed from cells that have not yet reached that final stage).?

Response: As described above and demonstrated in this manuscript, we conclude that one essential gene allele is sufficient for Leishmania parasite survival.

Comment: There are several other sections where the assumption is made that cells were null mutants but the actual status of the gene locus could/was not determined:

Figure S1. line 647-648. These data do not provide evidence that the gene is essential. Other explanations for the death of the cells cannot be excluded.

For all of these, the only claim that can be made is that the observed decay of the cells is consistent with an essential function of the gene.

Response: As we explained in our previous Response to reviewers, we had difficulty extracting the genomic DNA from the dying cell clumps for PCR analysis. However, we believe the disruption of all wild type essential gene alleles in those clones is the most likely cause of the cell dying and death as explained and illustrated in the section of essential genes (line 206-282) in the text using the AGC essential kinase 1 gene (LmxM.25.2340, AEK1) as a typical example (Fig. 2) to show how gene essentiality is normally determined with the rRNA-P stable protocol.

In addition, if the cell dying after cloning the double antibiotic resistance promastigotes into a 96 well plate was caused by the failed double strand DNA break repair, the promastigote would stop proliferating right after cloning and would not be able to form clumps containing more than ten to hundreds of cells; if the cell dying was caused by a spontaneous rare lethal mutation in the genome (unlikely), much fewer dying clones (maybe only one clone if it did happen) would be observed in a 96 well plate, instead of more than three dying clones regularly observed for disruption of an essential gene.

We understand the concern raised by this reviewer that despite the complete disruption of the essential gene alleles by the stable CRISPR gene targeting protocol is the most likely cause of cell death observed in the 96 well plate, due to difficulty to extract genomic DNA from the dying cell clumps, we did not have direct evidence so far to show that both the essential gene alleles in those dying promastigotes clumps were indeed disrupted by CRISPR with the antibiotic selection marker donor. However, it is important to point out that the DiCre inducibe gene deletion method, the most stringent method before CRISPR and recommended by this reviewer (see reference 2), also relies on observation of the cell death after complete deletion of the essential gene alleles to determine Leishmania gene essentiality. DiCre method also did not provide direct evidence (or definitive proof) to show that the floxed remaining essential gene allele in those dying Leishmania cells was indeed excised by the DiCre after addition of rapamycin.

To identify potential new drug targets, it is necessary to clearly determine the essentiality for all Leishmania genes [2]. The T7 transient system currently employs two methods to determine whether a gene is essential for Leishmania viability. Method 1 is to see, like the traditional gene targeting, if it is possible to delete the two chromosomal copies of the essential gene of interest by providing the cell with an extrachromosomal copy of the gene [2, 21,22]; Method 2 is to repeat the targeting attempt for a minimum of three times, if the null gene mutant could still not be generated with T7 system after three attempts, the gene can be considered as essential for Leishmania [21].Thus, both methods are labor intensive and time consuming. Method 2 at times could be not reliable as 4 to 6 attempts with T7 system were sometimes required to generate the null mutants for non-essential genes as shown in the L. donovani membrane protein gene study [22]. Nevertheless, as demonstrated in this study, using the rRNA-P stable protocol with single CRISPR plasmid and single antibiotic selection marker donor transfection, it was possible to determine gene essentiality by combining observation of the death of gene null mutant clones and detection of the WT essential gene band and the gene targeting band in all surviving clones in a 96 well plate [9,10,12,14]. 

Comment: line 45 (and 138-140) – “by directly observing gene null mutant promastigotes dying in culture”. The observation is: dying promastigotes. When these genes are targeted with the transient protocol, dying promastigotes are also observed. Whether or not the targeted gene is absent in these promastigotes is not known, in either case. All that can be concluded (in both protocols) is that failure to recover live cells is consistent with an essential function of the targeted gene (but not definitive proof).

Response: Although the complete sentence in line 43 should be “The rRNA-P stable protocol provides evidence for gene essentiality by directly observing null mutant promastigotes dying after cloning the antibiotic resistance promastigotes into a 96 well plate”, we believe line 43 (and 127-128) – “by directly observing gene null mutant promastigotes dying in culture” is appropriate in Abstract and Introduction because the Results section text will explain what it means the dying null promastigotes in culture after cloning into a 96 well plate. 

As we explained in our previous Response to reviewers, the dying promastigotes observed in transient protocol could be caused by several reasons (1) the non-transfected wild type cells killed by the two selecting antibiotics; (2) the cells killed by the two selecting antibiotics which were successfully transfected but no gene targeting took place; (3) the partially targeted (only one allele was targeted) cells killed by the second selecting antibiotic; (4) the cells with both alleles targeted by the same antibiotic selection marker donor, those cells would be killed by the another selecting antibiotic; (5) the cells with both alleles successfully targeted by each of the two antibiotic selection marker donors, dying and death of those cells after adding the two selecting antibiotics would be caused by the deletion of the targeting gene if it is essential. Thus, it is very difficult in the transient protocol to distinguish the cause of cell death between failure of the gene targeting and essentiality of a gene in such a large dying cell population. 

 In contrast, in the stable protocol, only one antibiotic selection marker donor is provided to the parasite. Before cloning into a 96 well plate, the promastigotes are already selected by the antibiotic specific to the selection marker donor, only cells with one or both alleles correctly targeted survive (if the gene is essential, the cells with both alleles disrupted will eventually die, see below). Because the stably expressed Cas9 and gRNA will continue to scan the genome and target the remaining WT gene allele after cloning into a 96 well plate, if the gene is essential for Leishmania viability, once the remaining WT allele in a cloned promastigote is targeted and disrupted, promastigotes in those wells would stop growing or multiply slowly to form clumps until the gene products (mRNA and protein) are diluted and degraded to the minimum level required for survival. Therefore, the slow dying of the cloned cells in the stable protocol could only result from the complete disruption of the essential gene alleles. In comparison, as described above, multiple factors could cause cell death in the transient protocol.

Comment: Figure S1. line 647-648. “Morphology of clumping dying clones (-/-) once the remaining copy of the gene has been disrupted by CRISPR.” The PCR shows that the WT band was present in all samples. No evidence is presented to show that the remaining copy of the gene has been disrupted. It is possible that this is what happens, likely even in many cases, but the data does not establish a direct link between the loss of the gene and the death of the cells. If the authors’ interpretation was correct, there should be a decrease of the WT PCR product over time. There is no evidence that the WT band decreases with time.

Respo

---

## [Editor Report · Decision Letter 2]

11 Dec 2024

Evidence for gene essentiality in Leishmania using CRISPR

PONE-D-24-31151R2

Dear Dr Matlashewski

We’re pleased to inform you that your manuscript has been judged scientifically suitable for publication and will be formally accepted for publication once it meets all outstanding technical requirements.

Kind regards,

Ben L. Kelly, Ph.D.

Academic Editor

PLOS ONE
---

## [Editor Report · Acceptance letter]

13 Dec 2024

PONE-D-24-31151R2 

PLOS ONE

Dear Dr. Zhang, 

I'm pleased to inform you that your manuscript has been deemed suitable for publication in PLOS ONE. Congratulations! Your manuscript is now being handed over to our production team.

Kind regards, 

on behalf of

Dr. Ben L. Kelly 

Academic Editor

PLOS ONE